# Pen-drawn Marangoni swimmer

Seo Woo Song [1,9,11] ✉, Sumin Lee[2,10,11], Jun Kyu Choe [3],
Amos Chungwon Lee[1,10], Kyoungseob Shin[2], Junwon Kang[4], Gyeongjun Kim[4],
Huiran Yeom [5], Yeongjae Choi[6], Sunghoon Kwon [1,2,7] ✉ & Jiyun Kim [3,8] ✉

Pen-drawing is an intuitive, convenient, and creative fabrication method for delivering emergent and adaptive design to real devices. To demonstrate the application of pen-drawing to robot construction, we developed pen-drawn Marangoni swimmers that perform complex programmed tasks using a simple and accessible manufacturing process. By simply drawing on substrates using ink-based Marangoni fuel, the swimmers demonstrate advanced robotic motions such as polygon and star-shaped trajectories, and navigate through maze. The versatility of pen-drawing allows the integration of the swimmers with time-varying substrates, enabling multi-step motion tasks such as cargo delivery and return to the original place. We believe that our pen-based approach will significantly expand the potential applications of miniaturized swimming robots and provide new opportunities for simple robotic implementations.

Pen has remained the most widely used tool for communicating and expressing ideas from ancient times to the current digital era, because of its versatility and high accessibility. Owing to its advantages, recent technological developments have enabled its usage as a fabrication tool to produce tangible and functional outputs. A pen has three primary benefits over other fabrication tools. First, it has extensive freedom of operation. It can easily be used on a broad range of substrates, including curved and bent surfaces and rigid and soft materials[1]. Hence, pen-drawn devices can be integrated with a wide variety of functional substrates. Second, the straightforward usability of the pen allows freestyle drawing and rapid prototyping of devices in a low-cost and highly efficient manner. Pen-drawing can readily deliver emergent and adaptive designs to devices; thus, it has been applied to the rapid prototyping of electronics[1,2] and microfluidic point-of-care testing devices[3,4]. The simple and accessible use of the pen can facilitate the onsite manufacturing of practical devices in resource-limited environments, as well as the application of new functional designs to the device. Third, various functions can be easily implemented by incorporating diverse functional materials into the ink solutions. For example, multilayer structures of display materials can be easily manufactured through sequential drawing using inks with different electronic components[5]. This capability of combining different functional inks can help integrate multiple functionalities into a single device. For the aforementioned reasons, the pen can be used for freestyle drawing of designs for various functions on diverse substrates, thereby extending pen-drawing into an intuitive manufacturing technique with high applicability. Owing to their various advantages, pens have been developed as versatile fabrication tools in many engineering applications such as microfluidics[3,4], flexible electronics[1,2,6], wearable devices[7,8], and 3D manufacturing[9,10]. These related studies have demonstrated the potential of pen-drawing as a simple and easy manufacturing technique for creative and practical applications.

The promising advantages of pen-based fabrication can be utilized to address the major challenges in the fabrication of swimming robots propelled by the Marangoni effect. The Marangoni swimmer is

¹Bio-MAX Institute, Seoul National University, Seoul, South Korea. ²Department of Electrical and Computer Engineering, Seoul National University, Seoul, South Korea. ³Department of Materials Science and Engineering, Ulsan National Institute of Science and Technology (UNIST), Ulsan, South Korea. ⁴Interdisciplinary Program for Bioengineering, Seoul National University, Seoul, South Korea. ⁵Division of Data Science, College of Information and Communication Technology, The University of Suwon, Hwaseong, South Korea. ⁶School of Materials Science and Engineering, Gwangju Institute of Science and Technology (GIST), Gwangju, South Korea. ⁷Inter-University Semiconductor Research Center, Seoul 08826, South Korea. ⁸Center for Multidimensional Programmable Matter, Ulsan National Institute of Science and Technology (UNIST), Ulsan, South Korea. ⁹Present address: Basic Science and Engineering Initiative, Children's Heart Center, Stanford University, Stanford, CA, USA. ¹⁰Present address: Meteor Biotech, Co. Ltd., Seoul, South Korea. ¹¹These authors contributed equally: Seo Woo Song, Sumin Lee. ✉e-mail: seowoo313@snu.ac.kr; skwon@snu.ac.kr; jiyunkim@unist.ac.kr

an aquatic robot that actuates on a liquid surface by releasing surface tension-lowering molecules around its body[11]. To move the robot as intended, it is necessary to fabricate elaborately designed engines containing fuel molecules that can generate the desired surface tension gradient[12]. The pen-based fabrication method can offer clear benefits for simple and rapid development of sophisticated engine designs. However, previously reported Marangoni swimmers had a tradeoff between motion complexity and manufacturing simplicity. Conventional Marangoni swimmers with simple designs have demonstrated limited motion capabilities such as linear or circular motion[13–17]. Recent studies have developed advanced engine systems to achieve complex motion control by using stimuli-responsive materials operated by external control systems such as light[18,19], heat[20], and magnetic field[21,22], or using a systematic design of the swimmer, such as the body shape[23,24], fuel-releasing profile[25,26], or fuel pattern[16,27–29]. However, a complex fabrication process is required to pattern stimuli-responsive materials or fuel-releasing components, which limits the practical applicability of Marangoni swimmers. Marangoni propulsion is a unique and powerful motion mechanism with a significantly high relative speed, considering the body length and

absence of a mechanical system[28]. The fabrication of Marangoni swimmers using pen-drawing will enable their rapid design and manufacturing, and allow them to be combined with various functional substrates to achieve highly advanced motion capability, thereby further expanding the applicability of this impactful motion mechanism.

This paper presents "pen-drawn Marangoni swimmers" for straightforward and freestyle patterning of fuel for Marangoni propulsion (Fig. 1a and Supplementary Movie 1). Camphor, a chemical that lowers the surface tension, induces a gradient in the surface tension around the vehicle and sublimates quickly, driving long-lasting motion[30]. We blended camphor into a polyvinyl butyral (PVB)-based ink to form a thin film of a camphor–PVB matrix, thus precisely controlling the release of the camphor molecules. The motion of the Marangoni swimmers can be programmed according to the drawing pattern and camphor concentration in the ink. As pen-drawing is the most convenient and intuitive method for designing diverse patterns[9], the proposed method can realize many different types of locomotion, including complex motion trajectories such as star- (Fig. 1b) or polygon-shaped. Another advantage of the proposed pen-drawn Marangoni swimmer is its versatile applicability to various

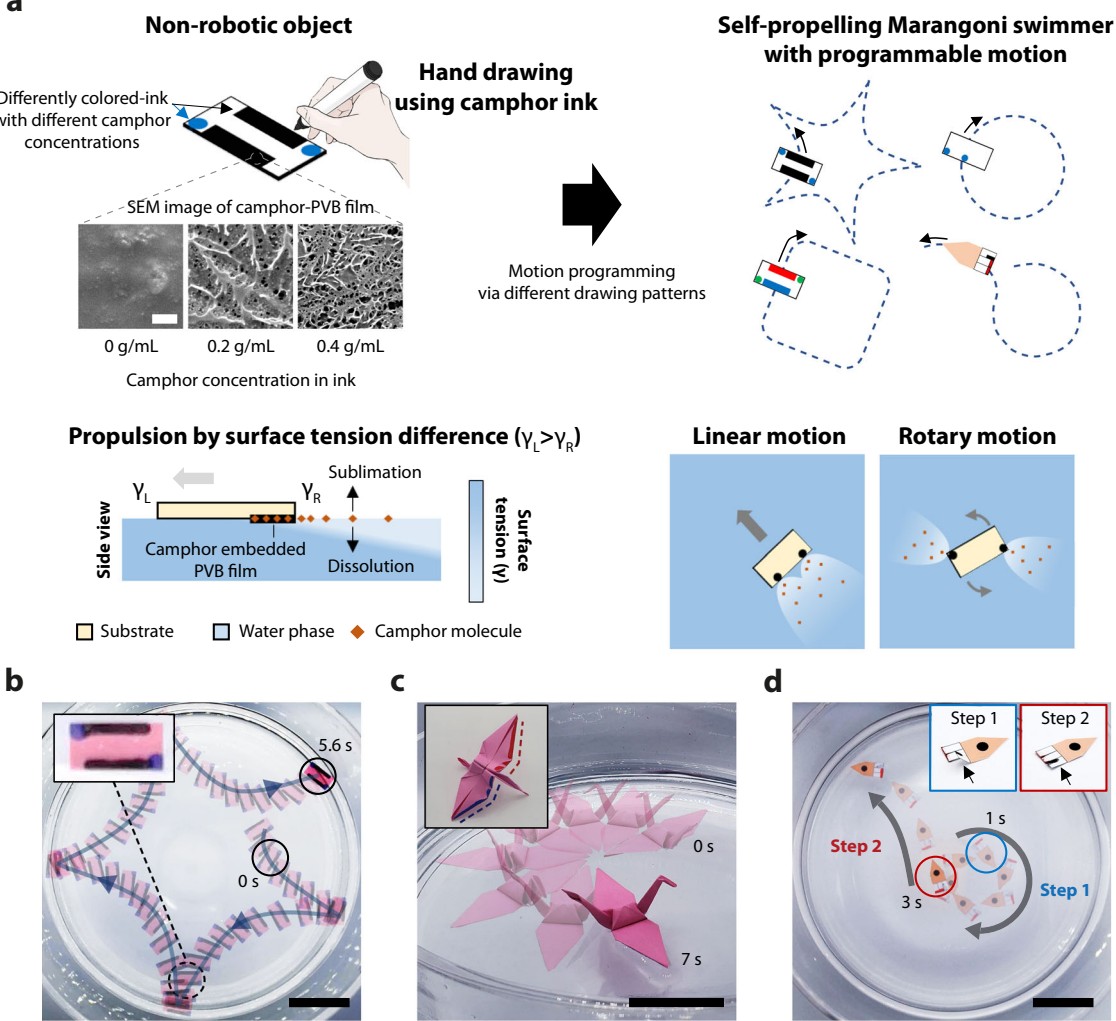

**Fig. 1 | Overview of pen-drawn Marangoni swimmers. a** Fabrication of pen-drawn Marangoni swimmers. The motion trajectory of the swimmers is determined according to the drawing pattern and concentration of the camphor ink. Different concentrations of camphor inks can be used together for programming a complex motion. **b** Complex motion trajectory programmed by a combination of different functional modules. **c** Versatile applicability of pen-drawn Marangoni swimmers on

diverse substrates. The pen-based approach enables patterning of camphor engines on a 3D structure. **d** Multistep motion programming by integrating a time-varying substrate. The folded paper is gradually unfolded when it is placed on the water surface. A camphor engine drawn on the folded region is activated when it touches the water surface. Scale bars: 1 μm (**a**) and 5 cm (**b**–**d**). SEM scanning electron microscopy, PVB polyvinyl butyral.

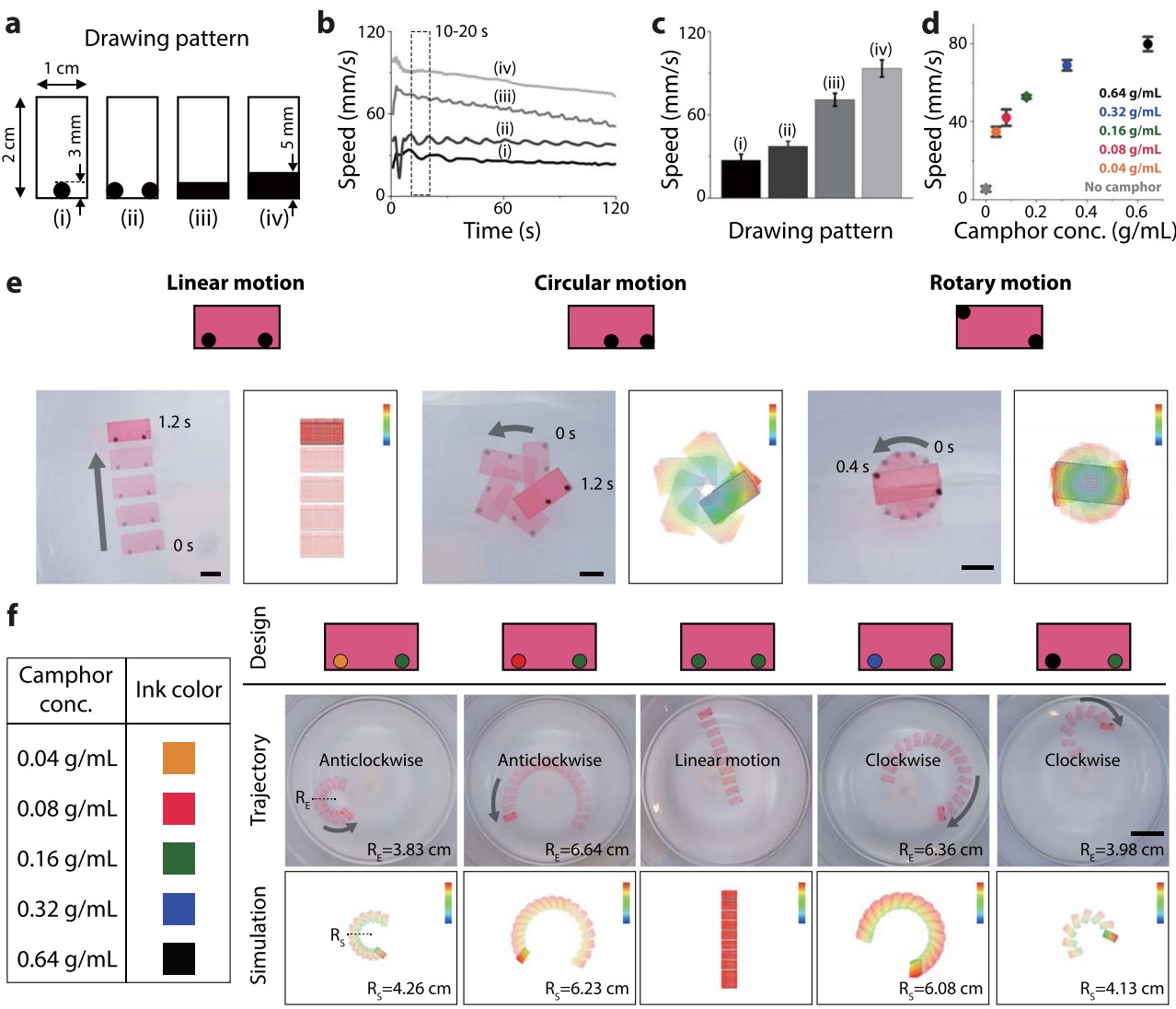

**Fig. 2 | Controllable motion of pen-drawn Marangoni swimmers. a–c** Speed of swimmer according to the drawing pattern. **a** Drawing pattern of camphor engine. **b** Speed of swimmers for different drawing patterns. **c** Average speed of swimmers between 10 s and 20 s ($n = 3$). **d** Speed of swimmers under different camphor concentrations. The same drawing pattern (iii) was used in this experiment ($n = 3$). Error bars represent standard deviation (SD). **e** Different motion trajectories according to the camphor engine arrangement and estimated motion trajectories simulated by FEA. Color bars represent the normalized speed based on the highest speed (red) and zero-speed (blue) in each simulation. **f** Motion programming using different concentrations of camphor inks. Scale bars: 1 cm (**e**) and 5 cm (**f**).

substrates, owing to the adaptability of pen-based fabrication[31,32]. We created Marangoni swimmers with various substrates, including plastic films and plates, 3D printed structures, paper origami (Fig. 1c), and even natural substrates such as plant leaves. Furthermore, we achieved time-dependent multistep motions by integrating pen-drawn Marangoni swimmers with substrates having time-varying structures (Fig. 1d). The ability to program diverse trajectories and multistep motions enables the practical implementation of cargo delivery functions, i.e., carrying the cargo to the target position and returning of the vehicle to the source position. Consequently, this study on the pen-based fabrication of highly programmable swimming robots lays the groundwork for simplifying robot manufacturing processes[33–36] and exploring the advanced motion capability and practical applicability of miniaturized swimming robots.

## Results

### Pen-drawn Marangoni swimmers with controllable motion
The locomotion of pen-drawn Marangoni swimmers can be determined based on the drawn pattern, concentration of fuel, and shape of the vehicle (Supplementary Fig. 4). To investigate the motion

controllability, we first measured the speed of the Marangoni swimmers according to the area covered with the camphor ink (Fig. 2a–c). As shown in Fig. 2a, four different patterns were drawn on a polyethylene terephthalate (PET) film (2 cm × 1 cm) using camphor ink with 0.32 g/mL camphor concentration. These swimmers moved straight, and their speeds were measured in a circular water tank (25 cm diameter) using ImageJ analysis (Supplementary Movie 2). As shown in Fig. 2b, the swimmers moved faster as the area covered by the camphor ink drawing increased. This was because a camphor–PVB film of larger area releases a greater amount of camphor, thereby inducing a steeper gradient in the surface tension. The speed of the swimmers gradually decreases owing to the consumption of fuel and saturation of camphor molecules at the water surface. The perturbation in the speed at the initial stage is due to the disturbance created when the swimmer is dropped on the water surface and the impact when the swimmer hits the water tank boundary. To exclude these perturbation effects from the quantitative analysis, we evaluated the average speed between 10 and 20 s (Fig. 2c). Using the same quantification method, we investigated the effect of the camphor concentration on the speed of the Marangoni swimmers for the same

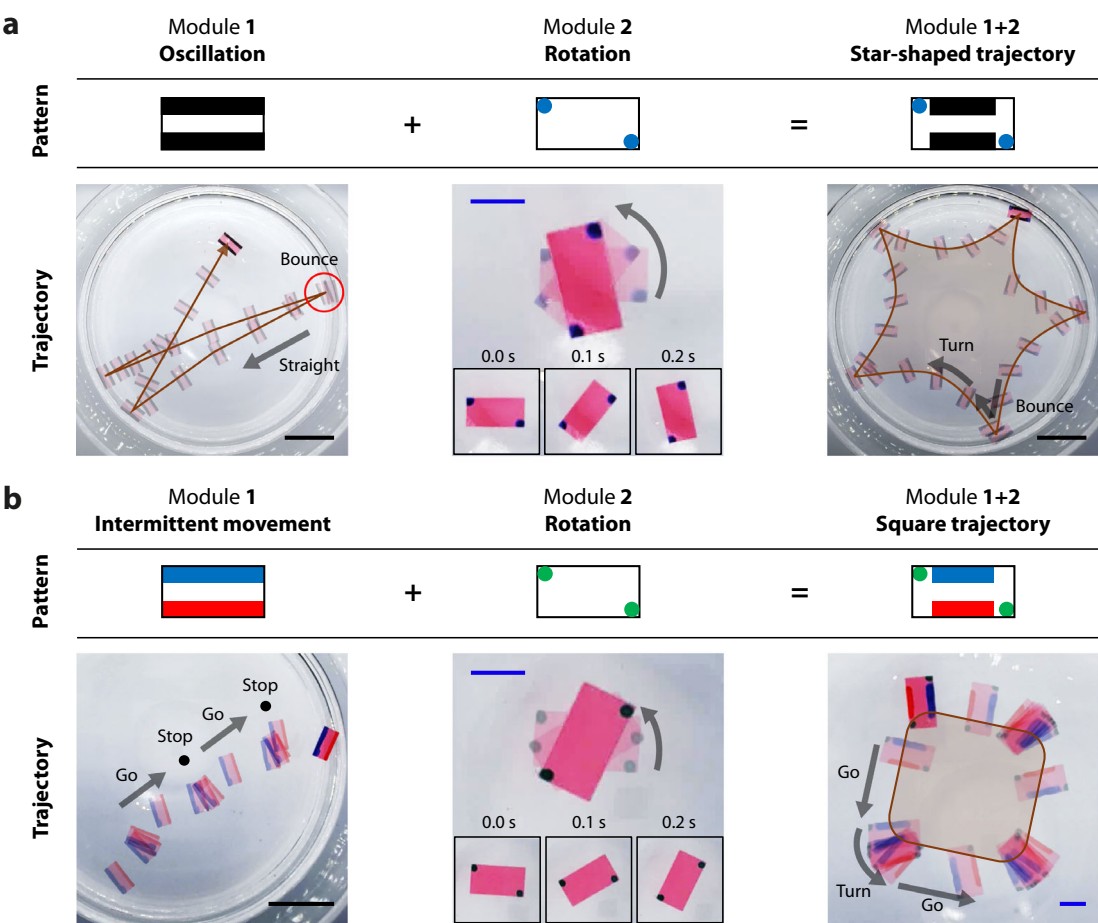

**Fig. 3 | Complex motion programming by combining different functional modules. a** Programming of a star-shaped trajectory. Unstable equilibrium resulting from symmetric camphor engines creates an oscillation motion. The oscillation module and rotation module are combined to realize a swimmer with a star-shaped trajectory. **b** Programming of a polygonal trajectory. Opposing camphor engines with different concentrations create an intermittent movement (see also Supplementary Fig. 2). A combination of intermittent movement module and rotation module results in a polygonal (square) trajectory. Scale bars: 1 cm (black) and 5 cm (blue).

drawing pattern (iii) (Fig. 2d). As expected, the higher the concentration of camphor in the ink, the faster was the speed of the swimmer. Thereafter, pigments of different colors were added to inks with different camphor concentrations to distinguish them from each other, as shown in Fig. 2d.

We show that the motion trajectory of a swimmer can be programmed using a drawing pattern (Fig. 2e and Supplementary Movie 3). We tested three different patterns: two camphor-ink dots on the vehicle but with different arrangements. Depending on the location of the dots (i.e., pattern of camphor-ink drawing), the swimmers have different trajectories, such as linear, circular, or rotational motion. The trajectory of the Marangoni swimmer can be mathematically modeled by calculating the driving force generated due to camphor release and assuming that the force acts on the camphor-ink-drawn area (referred to as the camphor engine hereinafter) (see Supplementary Text)[30,37–39]. However, due to the complexity of factoring in the swimmer's shape and the camphor engine design, we opted to implement this estimation through the finite element analysis (FEA) simulation (Fig. 2e) (see Supplementary Text).

This approach provides a brief understanding of how motion trajectories can be designed using drawing patterns. Subsequently, we investigated the variation in the motion trajectories when camphor inks of different concentrations were used to draw the same pattern (Fig. 2f and Supplementary Movie 3). As expected, the direction of rotation and radius of the trajectory changed according to the power of the left and right engines. The ease of manufacturing fuel engines

with different concentrations of the camphor ink in a single vehicle increases the design capability, thus enabling the simple programming of complex motion.

## Complex motion programming via a combination of different functional modules

Pen-drawn Marangoni swimmers have the advantage that the fabrication of the camphor engine is exceptionally simple, providing a wide range of design capabilities and motion programmability. We modularized the engines that performed each motion and easily combined these modules to perform more complex motions (Fig. 3 and Supplementary Movie 4). First, we demonstrated a star-shaped trajectory under a circular boundary condition by combining an oscillation module and a rotation module (Fig. 3a). Several studies have investigated swimmers with similar trajectories[40,41]; however, to the best of our knowledge, a design methodology based on this modular assembly concept has not yet been reported. Previous studies on camphor-based Marangoni swimmers have reported that the bilateral symmetric structures of the Marangoni swimmers have unstable equilibrium; hence, a small perturbation makes the swimmer move in a specific direction[39,42,43]. If such a symmetric swimmer moves in a water tank as a boundary condition, the collision with the wall causes a perturbation that causes the swimmer to move in the opposite direction, thereby performing an oscillatory motion that changes direction each time it collides with the wall (Fig. 3a, left). When combined with the rotation module (Fig. 3a, middle), the resulting swimmer moves in

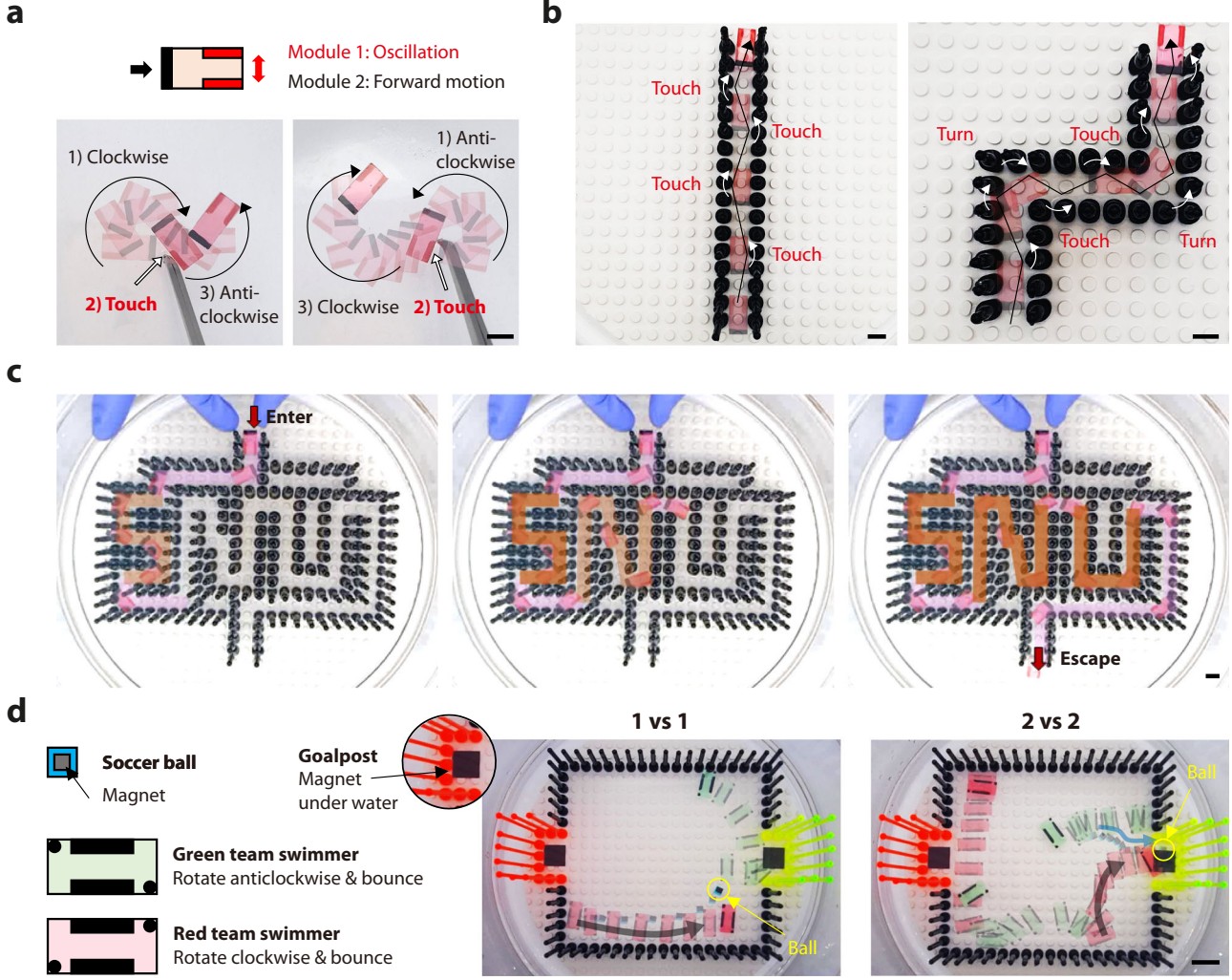

**Fig. 4 | Practical applications of functional module combination for complex motion programming. a–c** Path-finding swimming robot. **a** The black engine generates dominant propelling power to move forward. Red engine is a steering actuator to change directions when encountering obstacles. **b** Combination of these two modules enables the swimmers to navigate a narrow winding path. **c** Demonstration of escape from the SNU-shaped maze. **d** Robot soccer on a water surface with programmable Marangoni swimmers. Scale bars: 1 cm.

a circular trajectory in free space, and bounces and moves in the opposite direction when it hits a wall. In a water tank with a circular boundary condition, the swimmer repeats such movements, thereby drawing a star-shaped trajectory (Fig. 3a, right). Next, we achieved a polygonal (square) trajectory by combining an intermittent moving module and a rotating module (Fig. 3b). The periodic motion of camphor swimmers has been investigated in previous studies[42,44,45]. Similarly, we implemented the periodic movement of a pen-drawn Marangoni swimmer by creating an asymmetric design using camphor inks of two different concentrations (Fig. 3b, left). Our theoretical model of this periodic movement is shown in Supplementary Fig. 5. We combined the intermittent moving module with the rotation module (Fig. 3b, middle), and the resulting swimmer repeated the intermittent forward movement, followed by rotation in place. The rotation angle between two straight trajectories is affected by varying the concentration of the camphor ink. We demonstrated a square-shaped trajectory by selecting the optimal concentration of the camphor ink (Fig. 3b, right). We investigated the reproducibility of these complex motion programming, as shown in Supplementary Fig. 6. First, for the programming of the star-shaped trajectory, all five attempts from the five trials were successful in drawing star-shaped trajectories that traveled along the boundary while bouncing off a wall (Supplementary Fig. 3a). However, there were some differences in the angle between

each vertex depending on the attempt, which seems to be due to the inaccuracy caused by manual hand drawing. Three out of five attempts showed a five-pointed star shape, and the other two had approximately a four-pointed star shape. Secondly, the reproducibility of the polygonal trajectory was investigated as shown in Supplementary Fig. 3b. Every experiment from five attempts showed a polygonal trajectory with periodic movement, but one of them had a lack of rotation angle in the vertex, which led to a hexagonal path, rather than a square path like the others. We believe that this inaccuracy can be improved if we adopt a more precise printing method rather than manual hand drawing.

A combination of functional modules can be used to implement complex motions for practical applications (Fig. 4 and Supplementary Movie 5). First, we demonstrated a path-finding swimming robot to find a path out of a maze by applying an oscillation module at the head part and a forward-movement module at the tail part (Fig. 4a–c). A high concentration of the camphor ink (black, 0.64 mg/mL) is used in the tail part to create a dominant propelling power to move the swimmer forward, and the head part acts as a steering actuator to change directions when encountering obstacles (Fig. 4a). The underlying mechanism of the steering actuator is built on the oscillation module, as shown in Fig. 3a. Consequently, the swimmer can navigate a narrow path by changing directions through repeated collisions with the

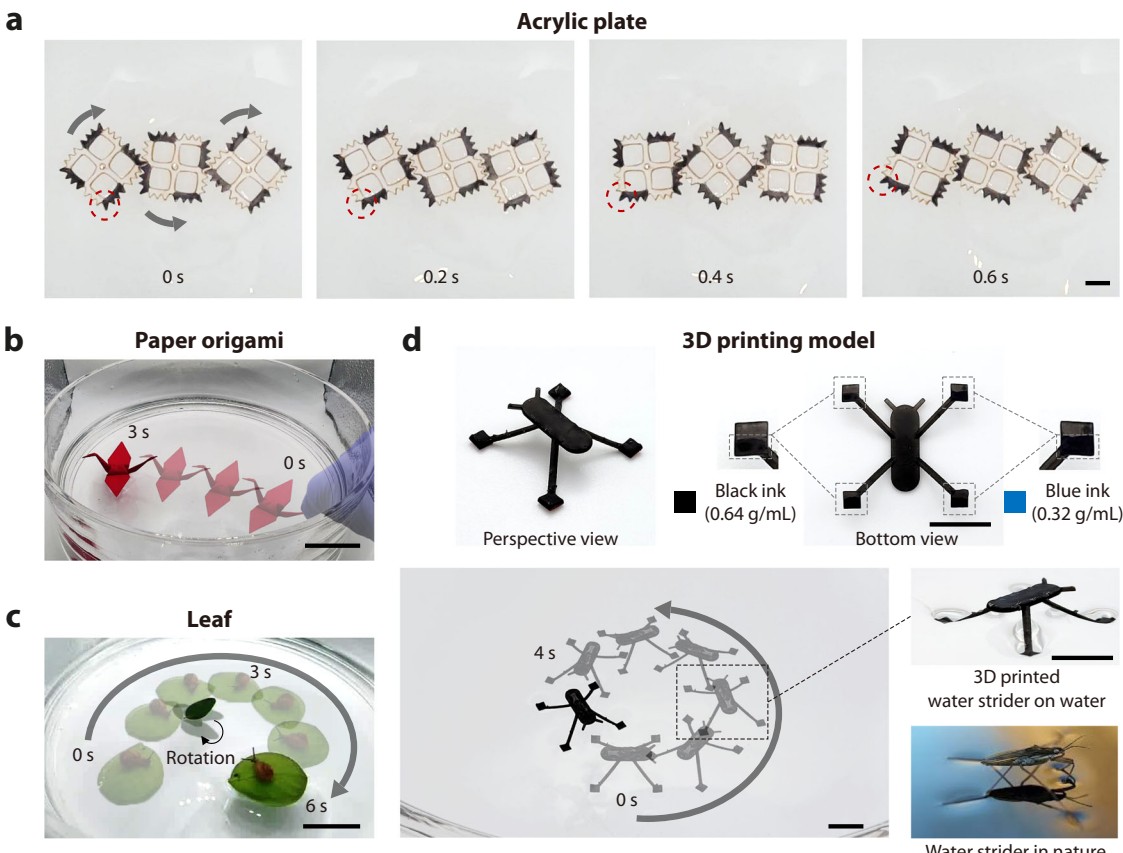

**Fig. 5 | Versatile applicability of pen-drawn Marangoni swimmers on various substrates.** The pen-based approach allows the fabrication of Marangoni swimmers with various substrates: **a** acrylic plate, **b** paper origami, **c** natural surfaces like leaves, and **d** 3D printed structures. Scale bars: 1 cm (**a**, **d**), 5 cm (**b**), and 3 cm (**c**). See also Supplementary Movie 5.

boundaries and can easily escape from the corners in either direction (Fig. 4b). As a result, the path-finding swimming robot can escape the "SNU"-shaped maze structure (Fig. 4c). Next, we demonstrated a swimming robot soccer game, which represents the interaction of multiple independently programmed swimmers (Fig. 4d and Supplementary Movie 5). The patterns of the swimmers engaged in the soccer game were designed to interact with each other while actively bounding the stadium boundary. The green and red swimmers were designed to rotate anticlockwise and clockwise, respectively. A small blue PET film with a magnet was used as a ball, which entered the goalpost area owing to magnetic attraction when the ball approached the goalpost. The self-propelling swimmers and the ball performed various soccer simulations such as dribbling and shooting via interactions with each other. Through the soccer game, we were able to simulate the interactions between multiple swimmers, for example, 1 vs. 1 and 2 vs. 2 games.

### Versatile applicability to various substrates and multistep motion programmability through integration with time-varying substrates

The proposed technology has great advantages owing to its versatile applicability to various substrates (Fig. 5 and Supplementary Movie 6). As reported in previous research, pen-based fabrication can be applied almost everywhere[9], enabling the "robotization" of non-robotic objects. The PVB-based ink forms a thin polymeric film after drying[9], which acts as a supportive membrane to maintain the camphor molecules on the drawn surface. Therefore, the camphor molecules are consistently released from the fabricated engine. We fabricated pen-drawn Marangoni swimmers with various substrates, including square gears made of acrylic plates (Fig. 5a), swimming crane made of paper origami (Fig. 5b), leaf boat transporting a live snail (Fig. 5c), and an artificial water strider fabricated using 3D printing (Fig. 5d). We also demonstrated the interaction between multiple swimmers fabricated using leaves (Supplementary Movie 7).

Multistep motion programming in a time-dependent manner is achieved when the pen-drawn Marangoni swimmer is implemented on substrates with time-varying structures (Fig. 6 and Supplementary Movies 8 and 9). It is well known that folded paper unfolds on a water surface as it absorbs water molecules (Fig. 6a). This phenomenon is mediated by the expansion of the compressed cellulose fiber network at the hinge region. Initially, the upper part does not contact the water but, eventually, contact is established as the paper becomes unfolded. Multistep motion programming was implemented by designing a camphor engine that is activated when the upper part contacts the water. We demonstrated three different two-step motion programming on a rocket-shaped vehicle (Fig. 6b, c and Supplementary Movie 8). Red camphor ink of lower concentration (0.08 g/mL) was used as the first-stage fuel and black camphor ink of higher concentration (0.64 mg/mL) as the second-stage fuel. Programmed motion differs according to the design of the engine arrangement. We observed a boost in the swimming speed when the second-stage fuel was applied in the midsection of a swimmer with symmetric structures (Fig. 6b). A change in the direction of the circular trajectory was observed when the application of the second-stage fuel was biased toward one side (Fig. 6c).

The time to transformation is determined by the initial folding angle. The larger the folding angle, the longer it takes the fiber matrix to expand to unfold the structure. By defining the transformation time

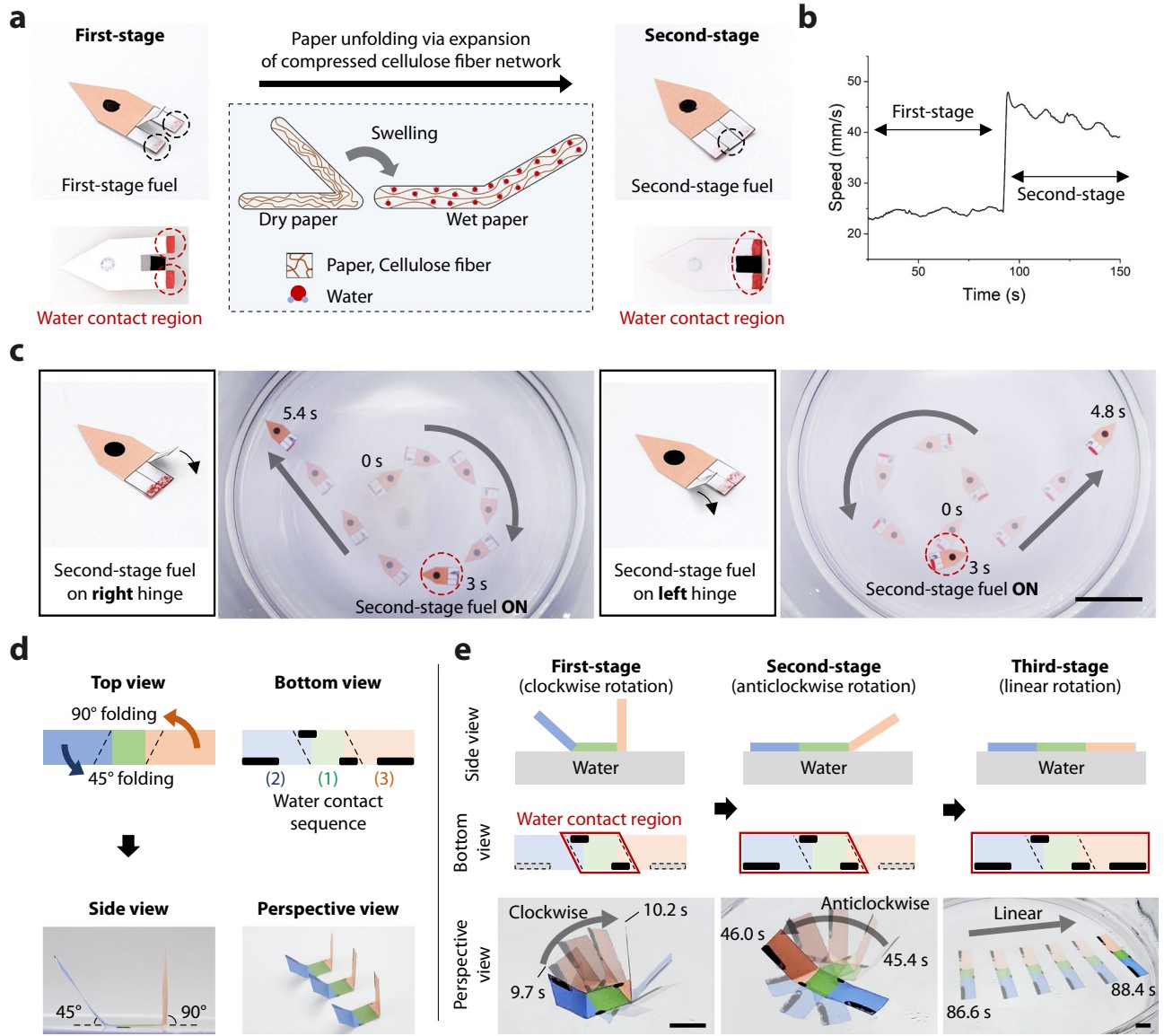

**Fig. 6 | Multistep motion programming on a time-dependent transformable substrate. a–c** Two-step motion programming of the rocket-shaped paper substrate. **a** The hinge of the folded paper is unfolded by the expansion of the compressed cellulose fiber when it touches the water surface. This phenomenon facilitates multistep activation of the camphor engines. **b** Speed of the swimmer is boosted as the second-stage engine is activated. **c** Different motion trajectories can

be programmed depending on the location of the second-stage engine. **d**, **e** Three-step motion programming via different folding angles. **d** Larger folding angle implies a longer time for unfolding and activation. Camphor engines in the green, blue, and orange areas are first, second, and third-stage engines, respectively. **e** The swimmer rotates clockwise, anticlockwise, and moves straight in the first, second, and third stages, respectively. Scale bars: 5 cm (**c**) and 1 cm (**e**).

as an additional controllable parameter, the swimmers can perform multistep motion involving more than two steps. We demonstrated a three-step swimmer, which can change its rotation direction and then move in a straight direction (Fig. 6d, e). The swimmer had two arms—the blue arm with a 45° folding angle, and the orange arm with a 90° folding angle (Fig. 6d). Consequently, the green part of the vehicle first touched the water, rotating the swimmer clockwise (Fig. 6e). Subsequently, as the blue and orange parts contacted the water surface sequentially, the motion of the swimmer changed to anticlockwise rotation and forward motion, respectively.

### Cargo delivery and return to the original place through multi-step motion
Pen-based engine fabrication allows easy integration of the Marangoni swimmers with various functional components. Multistep motion can be implemented not only through paper-folding transformation but

also by using other time-dependent substrates. Disassembly of a single swimmer into two individual swimmers was implemented using a water-soluble bridge (pullulan film) (Fig. 7a and Supplementary Movie 9). Two Marangoni swimmers with opposite structures were bonded together by using a water-soluble film. The swimmer had a bisymmetric structure and moved forward during the first stage. The bridging film began to dissolve when it contacted the water, and eventually separated the bonded swimmers. At the moment of disassembly, the residual bonding films acted as powerful propellants owing to the rapid diffusion of the pullulan molecules. After the subsidence of the instant boost in motion due to the water-soluble film, the individual swimmers followed the designated locomotion according to their camphor engine design and vehicle shape. Since the bridging film can decrease the surface tension of water as it dissolves and potentially acts as a Marangoni fuel, the location of the bridging films needs to be carefully designed to avoid any unwanted directional

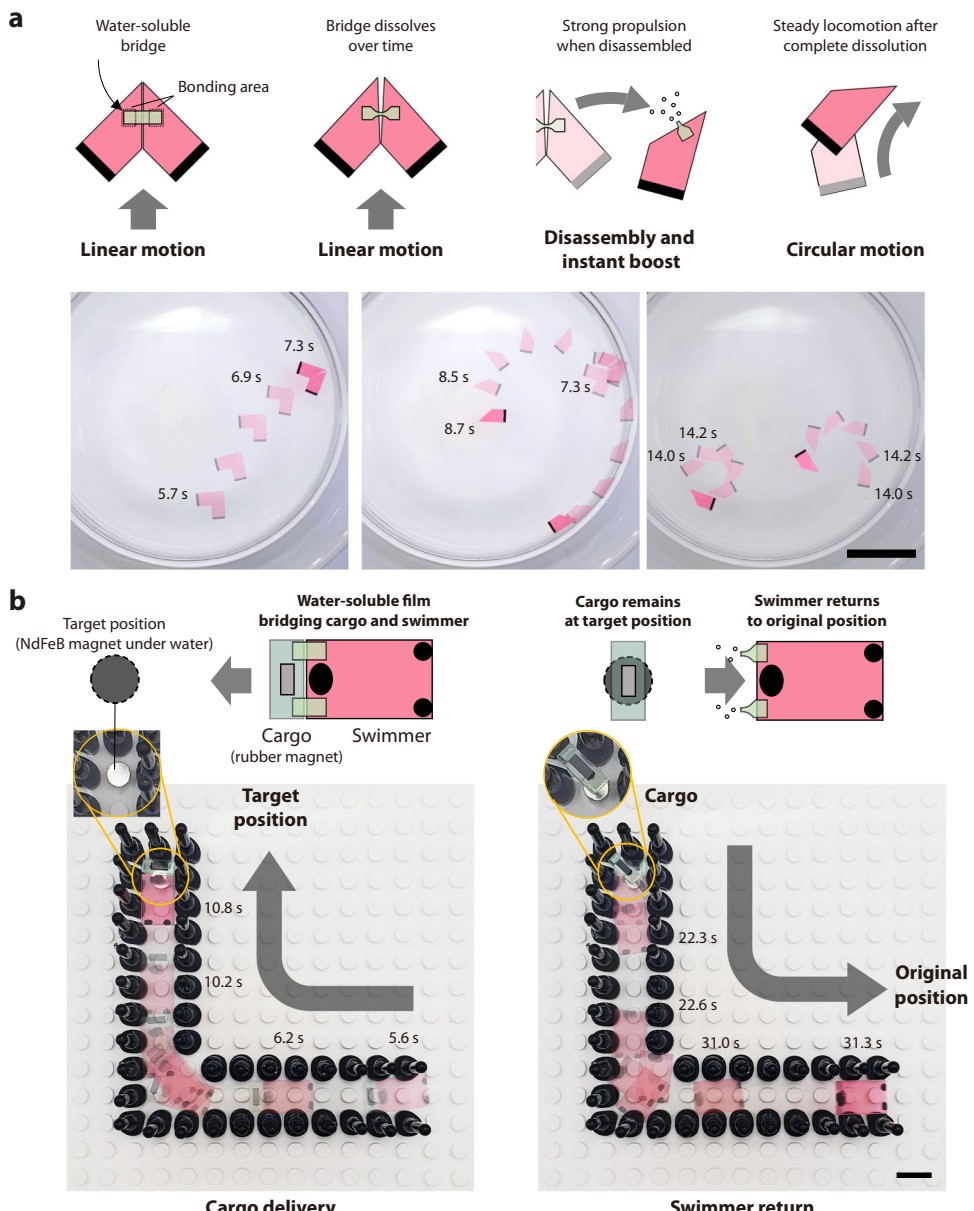

**Fig. 7 | Multistep motion programming using water-soluble bridges and its application in cargo delivery. a** Multistep motion programming by disassembly of the swimmers connected by a water-soluble bridge. **b** Cargo delivery and return using multistep motion programming. The swimmer returns to the original position after it releases the cargo at the target position. Scale bars: 5 cm (**a**) and 1 cm (**b**).

changes before disassembly. To achieve this, we designed the pullulan films to be bilaterally symmetrical around the heading direction and to not be exposed to the front or back of the vehicle. This makes that no rotational, forward, or backward propulsion force is generated by the bridging film before disassembly.

We could also adjust the time for disassembly by controlling the width of the bridging film, thereby realizing two-step disassembly and, consequently, three-step motion (Supplementary Fig. 7 and Supplementary Movie 9). By using multistep motion programmability with a water-soluble film, the swimmer could perform cargo delivery by reaching the target destination through a meandering path and returning to the starting point after releasing the cargo unit (Fig. 7b). The navigation mechanism was built on the swimmer shown in Fig. 4a. The time for cargo detachment was designed to maintain bonding until the swimmer reached the target position. Approximately 10 s after the swimmer reached the target position, the cargo and vehicle were disassembled. The cargo maintained the target position via a

magnetic attraction force, and the swimmer returned because of multistep motion programming.

## Discussion

We developed a Marangoni swimmer capable of highly complex and functional motion programmability by using a simple pen-drawn engine. Previously reported motion-controllable Marangoni swimmers mostly required bulky equipment to fabricate swimmers with designed fuel patterns or stimuli-responsive motor units. In this study, we adopted a pen-based approach to pattern camphor engines for Marangoni propulsion systems. This simple motor system, which is flexible in terms of the fuel composition, spatial engine pattern, and substrate, allows us to program complex motions and perform advanced robotic tasks.

Previous studies have revealed that the lifetime of a Marangoni swimmer is affected by various factors, including the initial fuel loading amount[28], fuel-releasing rate[21,46], and the saturation of the water

surface with surface tension-lowering molecules[47]. Although the lifetime of our Marangoni swimmer, ranging from several minutes to 10 min, is shorter than those reported in recent studies, it can be prolonged by adjusting ink composition[21,46]. Slowing down the camphor release rate by increasing the PVB concentration in ink to make the matrix denser or mixing additives to increase the matrix's hydrophobicity can be considered to prolong the swimmer's lifetime. However, it is important to note that this approach may decrease the power of the camphor engine, which can affect the swimmer's movement. Therefore, this tradeoff needs to be carefully considered when modifying the ink composition to increase the lifetime.

The intrinsic features of the pen-based approach provide several advantages. First, the versatile applicability of the pen-based approach to various substrates and its high accessibility not only widen its potential applications through integration with other functional substrates, but also enable multistep motion programmability by adopting transformable substrates. We demonstrated a multistep-motion programmable swimmer, its practical application for cargo delivery, and recovery of the robot in an inlet-only environment. We believe that this motion capability enables the use of miniaturized autonomous robots in environments where human access is restricted. Furthermore, simple implementation of dynamic engine designs will inspire studies on the kinetics of Marangoni swimmers with higher motion programmability.

Although the proposed hand drawing-based method offers exceptionally straightforward and highly accessible fabrication of Marangoni swimmers, the manual use of pen may limit the accuracy, reproducibility, and availability for miniaturization. However, the advantage of the proposed method is not just the ease of use. The ink-based printing method provides the advantage of making it easy to process different concentrations of fuel ink together on the same substrate. This enables highly advanced motion programmability. Furthermore, the ink-based fabrication of Marangoni swimmers is still compatible with conventional 2D printing technologies, such as penplotter and inkjet printing[9]. Combined with computer-aided design (CAD) and automated printing, more accurate, miniaturized, and mass-producible fabrication is achievable. We anticipate that even the general public or young students without subject expertise can study the Marangoni propulsion robot as a part of the STEM (Science, Technology, Engineering, & Mathematics) curriculum because the main components of camphor ink are non-toxic materials widely used in everyday life.

## Methods

### Preparation of camphor ink and pens

The camphor ink used in this study was developed based on a commercial dry erase marker[48]. Typically, the dry erase marker ink generates a thin PVB film after drying. Because the PVB film is not soluble in water, it acts as a supporting membrane that holds the camphor molecules on the ink-drawn surface. The camphor ink was blended using the following protocol. First, 7 wt% PVB, 70 wt% solvent (ethanol: isopropyl alcohol = 8:2), 15 wt% plasticizer (dibutyl sebacate), and 5 wt% pigment were mixed together, and the mixture was vortexed at a speed of 1000 rpm for 24 h to obtain a homogeneous solution. Subsequently, the designated amount of camphor was added to the mixture. The color of the pigments was determined according to the camphor concentration, as described in Fig. 2f, unless otherwise noted. The final mixture was filled into an empty pen (Green Board, Green B&T, Korea) with a 3 mm nib.

PVB, dibutyl sebacate, and camphor were purchased from Sigma-Aldrich. The orange, red, green, blue, and black pigments used in this study were pigment orange 5, pigment red 254, pigment green 7, pigment blue 15, and pigment black 5 from TCI Chemicals, respectively. All these pigments are water-insoluble; hence, they were not dissolved in water when the swimmers moved on the water surface.

### Fabrication of pen-drawn Marangoni swimmers

The swimmer bodies shown in Figs. 1b, 2, and 7 were made by cutting a PET film (Daejin Education, South Korea) using a $CO_2$ laser cutting machine (Rexbot L3020, South Korea). Commercialized colored origami paper was used in the experiments shown in Figs. 1c and 5b. Double A printing papers were used in the experiments shown in Figs. 1d and 6. Acrylic plates of 3 mm thickness were used in the experiment shown in Fig. 5a. A Form 3 resin 3D printer (Formlabs) was used to print the 3D water strider model shown in Fig. 5d. Pullulan film <Cool Mint Listerine Pocketpaks> was utilized as a water-soluble bridging film. The pullulan film was cut to an appropriate width to meet the designated disassembly time, and the film was bonded to the PET films using a cyanoacrylate adhesive. Finally, the camphor engines were installed on various substrates using the pen-drawing method with camphor-ink pens.

### Simulation of swimmers' locomotion

Finite element simulation was implemented using ABAQUS software. To simplify the simulation model for estimating the trajectories of the swimmers, we considered the driving and drag forces acting on the swimmer on a smooth wave-free surface. The driving force of the swimmer was calculated using equations based on previous research on camphor-based Marangoni swimmers[30]. The frictional coefficient μ was calculated based on the data shown in Supplementary Text, and the driving force according to the fuel concentration was calculated using μ and the data shown in Fig. 2d. To account for the applied area of the camphor ink, we first hypothesized that the driving force of the camphor engine is directly proportional to its area in a local region, and then converted the driving force into driving pressure by dividing the driving force by the working area. Similarly, for the drag force of the swimmer on a fluid surface, we applied surface-based stagnation pressure loads to the swimmer's body[49]. The constant value for the stagnation pressure model was fitted based on the circular motion data, as shown in Fig. 2e. Finally, the simulation model with defined parameter values was applied to predict the trajectories of other swimmers, as shown in Fig. 2e, f, by using the adjusted area and intensity of the driving pressure. Additional information is provided in the Supplementary Text.

## Data availability

All data generated or analyzed during this study are included in the published article and its Supplementary Information and are available from the corresponding author on request.

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

## Acknowledgements

This research was supported by: National Research Foundation of Korea (NRF) grant 2020R1C1C1007665 (S.W.S.), 2017K1A4A3015437 (J.Kim), NRF-2020R1A2C2102842 (J.Kim), NRF-2021R1A4A3033149 (J.Kim), NRF-2020R1A3B3079653 (S.K.), NRF-2022M3C1A3081366 (Y.C.), and 2019-Global PhD. Fellowship Program (S.L. and J.K.C.), the BK21 FOUR program of the Education and Research Program for Future ICT Pioneers grant (S.K.), the Fundamental Research Program of the Korea Institute of Material Science (PNK7630) (J.Kim), and the Research fund of UNIST (1.220052.01) (J.Kim).

## Author contributions

S.W.S., S.L., J.K.C., and Y.C., designed experiments. S.W.S., S.L., K.S., and H.Y. obtained experimental results. S.W.S., S.L., J.K.C., and G.K. investigated theoretical backgrounds. S.W.S., S.L., J.K.C., and J.Kim designed and obtained simulation results. S.W.S., S.L., J.K.C., J.Kang, Y.C., S.K., and J.Kim designed and illustrated figures. S.W.S., S.L., J.K.C., A.C.L., G.K., H.Y., S.K., and J.Kim. wrote a manuscript. S.W.S., S.K., and J.Kim supervised the project. All authors contributed to revising the paper.

## Competing interests

The authors declare no competing interests.
