## [Peer Review File · Nature Communications]

REVIEWER COMMENTS

Reviewer #1 (Remarks to the Author):

The authors of this article present a series of compelling experiments on Marangoni swimmers created in an extremely versatile manner employing direct drawing of patches releasing camphor using a pen.

The physics of these systems has been extensively explored and multiple applications have been already proposed. However, the simplicity and versatility of the design proposed here make this manuscript a very compelling and interesting read. Given the simplicity of the approach, I expect many to follow these steps. The visual information in the form of figures and supporting videos is extremely clear and the text is easy to follow also for non-experts.

Even if the experiments and their interpretation are simple, they are carefully carried out and are supported by detailed calculations.

I therefore recommend publication in Nature Communications. I only have a minor comment, which I would like the authors to consider before publication.

- The equations on page 6 are just placed there and are not really connected to the rest of the text. I suggest that they are moved to the SI, where the authors describe the modelling in detail.

Reviewer #2 (Remarks to the Author):

The authors describe a technique for producing Marangoni swimmers and other objects that generate Marangoni flows. The technique is based on loading a marker pen with ink that contains camphor. Using the pen to write on otherwise generic objects patterns them with local stores of camphor that are gradually released and cause Marangoni flows when the surface contacts water. The authors demonstrate this with pieces of PET film, acrylic, paper, leaves, and 3d printed structures. Moreover, the authors demonstrate that such Marangoni swimmers can be designed to move linearly, spin around, or perform more complex manoeuvres, such as following a maze-like confined path by combining a pattern that generates forward propulsion with a pattern that causes side-to-side oscillations.

To add further complexity, the authors used folded paper, which gradually unfolds upon contact with water to expose different patterns to the water. Different conformations of the structure could be associated with different Marangoni-driven motion.

The novelty in this work is the pen-drawing technique of patterning objects for Marangoni flow. This is significant because it allows very quick and convenient production of swimmers with different patterns, which could facilitate further detailed studies.

The authors have shown a good variety of examples from simple to complex, time-dependent behaviours. In my opinion, the work is of good quality and sufficiently impactful to warrant consideration for publication in Nature Communications. The manuscript is clearly written and figures are well made. Experimental procedures appear to be sound and adequately described, but this aspect is beyond my expertise.

The theoretical modelling and simulation results, however, were somewhat unclear and weak. From the descriptions in the main text and supplementary text, the friction coefficient μ and the driving force are inputs for the finite element analysis. It is not clear what the purpose of the FEA is, then. Mechanical properties of the material are quoted, so presumably strains are calculated but they are not discussed at all in the text. It seems that FEA was only used to calculate trajectories of the swimmers but this approach is not necessary if one already has a formula for the driving force. One can directly solve the equations of motion (2) and (4).

The principles that are demonstrated in this work are interesting but there is scope for better quantitative modelling of the motions produced by different patterns. The theoretical analysis in the supplementary text attempts to address this but appears to have some significant flaws. For instance, it is assumed that the direction of the forces at the patterned corners is vertical (as shown in the supplementary figure) but no explanation is given for this. Moreover, in the process of deriving the magnitude of these forces, it is stated that (i) the spatial gradient of camphor is negligible and that (ii) the concentration of released camphor molecules is negligible. Each of these assumptions should mean that the surface tension is the same everywhere, i.e., there is no Marangoni flow and no motion of the swimmer. Even with the presentation clarified, the basis of this model is the formula for the force proposed in Ref 27, which was based on linear motion and therefore had an unambiguous direction even if details of the (symmetric) shape were omitted. For rotational motion, the direction of the forces each ink patch produces must depend on the shape of the swimmer.

Other comments and questions for the authors:

1. In equation 2 of the main text and equation 3 of the supplementary material, r and r_0 should be γ and γ_0 , respectively. Also, in the supplementary text, equations 6-7 have missed the parentheses around $\Gamma(x,t)$, making it look like a product of γ and Γ whereas actually Γ is the argument of γ .

2. "The mean surface concentration can be derived using equation (3)... This implies that Marangoni propulsion, which is the difference in the surface tension around the vehicle, increases when the release ratio of the camphor molecules increases." It is reasonable that the propulsion increases when the release ratio increases but this is not related to equation (3), which only describes the mean concentration (which does not contribute to Marangoni propulsion).

3. Variables in the equations should be defined. Also, the mean surface concentration in equation (3) should be distinguished from the spatially-dependent variable, e.g., with a bar above the Γ symbol.

4. Details of the FEA were missing, such as shape and dimensions of the 3D object. What direction is the driving pressure applied for the dots? Following figures in the manuscript, the dots should be placed on the horizontal surfaces so pressure would act vertically and not drive horizontal motion.

5. "We demonstrated a square-shaped trajectory by selecting the optimal concentration of the camphor ink (Fig 3b, right)." This suggests that the camphor concentration is the main parameter that needs to be tuned but it must surely also be somewhat sensitive to how the pattern is drawn, e.g., thickness and lengths of lines, radius of dots. More generally, could the authors comment on how reproducible the behaviour of swimmers was? Is it possible to reliably make swimmers with the same motion? Closed, periodic trajectories such as the square and star-shaped paths should be especially difficult to control accurately.

6. The authors mention that the dissolution of the water-soluble bridge causes a boost in motion as the bonded swimmers separate. How does the film affect motion before separation? Can the film itself be used as a Marangoni fuel? Were any control cases tested for initially bonded swimmers where no camphor ink is used?

Reviewer #3 (Remarks to the Author):

The paper reports on a new experimental system on "Marangoni swimmer". A material that releases surface active chemicals to the surrounding water surface is known to exhibit self propulsion. Camphor is one of the most famous chemicals for such motion. When a camphor particle is put on the water surface, it moves spontaneously since surface tension around the particle decreases due to the camphor molecules and the surface tension working on the particle becomes unbalanced. The authors newly designed the material that shows the self-propulsion. The advantage of the material is the ease in molding. The authors claim that they can make the swimmer like "pen-writing". Then, they have shown many examples on the interesting behaviors using the new material. The development of the material is novel and important, and therefore I think the manuscript has a potential to be published in Nature Communications. However, each result shown in the figures and movies has been reported using the other systems. Thus, the authors have to comment on the previous related results. Moreover, the theoretical and numerical parts are poorly written. The manuscript should be reviewed again after the author revises based on the comments.

1) The theoretical parts are quite unclear. The authors briefly show the equations 1 to 3. However, they do not clearly show what are the variable and what are the constants. Especially, in equations 1 and 2, $\Gamma(x,t)$ looks like a variable, but in equation 3 $\Gamma(t)$ is given. It makes the readers confused.

2) As for the theoretical work of the camphor particles with various shapes has been reported. The author did not cite the previous work on the theoretical framework. The followings are some of them:

- Mathematical modeling and analysis

M. Nagayama, S. Nakata, Y. Doi, and Y. Hayashima, A theoretical and experimental study on the unidirectional motion of a camphor disk, *Physica D* 194, 151 (2004).

- Mathematical modeling and analysis for two-dimensional system

H. Kitahata, K. Iida and M. Nagayama, Spontaneous motion of an elliptic camphor particle, Phys. Rev. E, 87, 010901 (2013).

- Review for the camphor particle motion

S. Nakata, M. Nagayama, H. Kitahata, N. J. Suematsu, and T. Hasegawa, Physicochemical design and analysis of self-propelled objects that are characteristically sensitive to environments, Phys. Chem. Chem. Phys. 17, 10326

(2015)

3) In the supporting information, the authors show the mathematical model for the two-dimensional motion. I cannot totally understand the model in SI. Since the authors consider the two-dimensional motion, I think $\Gamma(x,t)$ should be $\Gamma(x,y,t)$. Thus, the expression $\Gamma(x,t)$ in equation (3) in the main text should no longer work.

4) Related to the comment 3, the force originating from the surface tension should work the whole object. I do not think this fact is not included in the model.

5) The authors fails to cite the references that are much related to the present contents:

- Translational and Rotational motion of a camphor particle

S. Nakata, Y. Iguchi, S. Ose, M. Kuboyama, T. Ishii and K. Yoshikawa, Self-Rotation of a Camphor Scraping on Water: New Insight into the Old Problem, Langmuir, 1997, 13, 4454–4458.

- Design of the self-propelled objects using camphor particles

H. Morohashi, M. Imai, and T. Toyota, Construction of a chemical motor-movable frame assembly based on camphor grains using water-floating 3d-printed models, Chem. Phys. Lett. 721, 104 (2019)

- New material for self-propulsion including polymers

R. J. G. Löffler, M. M. Hanczyc and J. Gorecki, A Perfect Plastic Material for Studies on Self-Propelled Motion on the Water Surface, *Molecules*, 26, 3116 (2021).

- Interesting trajectory of self-propulsion

H. Kitahata, Y. Koyano, R. J. G. Löffler, and J. Górecki, Complexity and bifurcations in the motion of a self-propelled rectangle confined in a circular water chamber, *Phys. Chem. Chem. Phys.* 24, 20326-20335 (2022).

S. Tanaka, Y. Sogabe, and S. Nakata, Spontaneous change in trajectory patterns of a self-propelled oil droplet at the air-surfactant solution interface, *Phys. Rev. E* 91, 032406 (2015).

Reviewer #4 (Remarks to the Author):

Marangoni swimmers are designed by drawing with a pen with camphor ink. Camphor release induces surface tension gradients which, in turn, causes swimmer propulsion. The authors demonstrate swimmers that can follow complex trajectories and successfully execute multistep tasks. The idea and implementation very interesting and can inspire further design and prototyping of multifunctional swimmers. The paper is well written and organized and will be of interest to a broad readership. Several changes can be suggested before publication:

1. In Figure 2 the legend in panel d need to be arranged concentration. In the same figure, clarify what the colorbars in panels e and f represents.

2. An explain of how intermittent motion is achieved need to be included.
3. Comment on the reproducibility of the results, especially when complex trajectories are considered.
4. It will be useful to discuss miniaturization of the technology and relevant limitations.
5. Discuss what sets the overall time of the camphor release and how the release time can be regulated.

Response to reviewers' comments for the manuscript – “Pen-drawn Marangoni swimmer”

General Response

We appreciate the thoughtful comments from the reviewers based on the manuscript. We find their feedback very helpful in improving and completing our manuscript. In this letter, we address the reviewers' comments. All revised content in response to the comments is **highlighted in yellow** in both the Manuscript and the Supplementary Information. Text taken from the Manuscript or Supplementary Information is indicated in *italics* in this Response Letter.

Response to reviewer #1's comments

General Comment: The authors of this article present a series of compelling experiments on Marangoni swimmers created in an extremely versatile manner employing direct drawing of patches releasing camphor using a pen. The physics of these systems has been extensively explored and multiple applications have been already proposed. However, the simplicity and versatility of the design proposed here make this manuscript a very compelling and interesting read. Given the simplicity of the approach, I expect many to follow these steps. The visual information in the form of figures and supporting videos is extremely clear and the text is easy to follow also for non-experts. Even if the experiments and their interpretation are simple, they are carefully carried out and are supported by detailed calculations. I therefore recommend publication in Nature Communications. I only have a minor comment, which I would like the authors to consider before publication.

Comment 1: The equations on page 6 are just placed there and are not really connected to the rest of the text. I suggest that they are moved to the SI, where the authors describe the modeling in detail.

Response for General Comment and Comment 1: Thank you for the supportive and positive feedback on our manuscript. We have carefully addressed the minor comments related to the equations by moving them to the Supplementary Information section and providing improved descriptions to enhance reader comprehension. The relocated equations can be found highlighted in yellow in the revised Manuscript and Supplementary Information.

Manuscript

We show that the motion trajectory of a swimmer can be programmed using a drawing pattern (Fig. 2e and Supplementary Movie 3). We tested three different patterns: two camphor-ink dots on the vehicle but with different arrangements. Depending on the location of the dots (i.e., pattern of camphor-ink drawing), the swimmers have different trajectories, such as linear, circular, or rotational motion. The trajectory of the Marangoni swimmer can be mathematically modeled by calculating the driving force generated due to camphor release and assuming that the force acts on the camphor-ink-drawn area (referred to as the camphor engine hereinafter) (See Supplementary Text)^{30,37-39}. However, due to the complexity of factoring in the swimmer's shape and the camphor engine design, we opted to implement this estimation through the finite element analysis (FEA) simulation (Fig. 2e) (see Supplementary Text).

Supplementary information

Mathematical modeling and FEA simulation for predicting camphor-driven swimmer trajectories

Table. Parameters for mathematical modeling and FEA simulation

Notation	Description	Unit
$\Gamma(x, y, t),$ $\Gamma(x, t), \bar{\Gamma}$	Surface camphor concentration Notation depends on modeling dimensions	$[\text{mol m}^{-2}]$
$f(x, y)$	Supply of the camphor molecules from the camphor engine to the water surface (in 2-dimension modeling)	$[\text{mol s}^{-1}\text{m}^{-1}]$
γ	Surface tension	$[\text{N m}^{-1}]$
γ_0	Surface tension of pure water	$[\text{N m}^{-1}]$
p	Positive constant	
$F_{\text{Driving force}}$	Driving force of the moving camphor object (in 2-dimensional modeling)	$[\text{N}]$
F_{drag}	Drag force exerted on the object (in 2-dimensional modeling)	$[\text{N}]$
D	Diffusion constant	$[\text{m}^2 \text{s}^{-1}]$
k	Sum of k_s and k_d	$[\text{s}^{-1}]$
k_s	Sublimation rate from the water surface	$[\text{s}^{-1}]$
k_d	Dissolution rate from the water surface	$[\text{s}^{-1}]$
x_0	Position at the edge of the camphor engine	$[\text{m}]$
m	Mass of the camphor boat	$[\text{kg}]$

μ	Friction coefficient	$[N\ m^{-1}\ s]$
F_w	Driving force of the moving camphor boat (in 1-dimensional modeling)	$[N]$
F_{w0}	Driving force of the stationary camphor boat (in 1-dimensional modeling)	$[N]$
L	Length of the contact line between the camphor disk and water surface	$[m]$
v_{steady}	Steady state velocity of camphor engine	$[m\ s^{-1}]$
α	Supply rate of camphor molecules from the camphor engine to the water surface (in 1-dimensional modeling)	$[mol\ s^{-1}]$

Predicting an object's trajectory can be achieved through a comprehensive analysis of the net force and net torque applied to the object. By understanding these factors, it becomes possible to anticipate the motion and behavior of the Marangoni swimmer in various situations.

For example, let's consider a rectangular Marangoni swimmer with two camphor engines placed on one edge, as shown below.

We refer to the two-dimensional modeling of a Marangoni swimmer's motion studied by H. Kitahata and colleagues¹. The surface concentration of camphor molecules is defined as $\Gamma(x, y, t)$, which is represented as Γ for simple notation in equation (1). To figure out the trajectories of the swimmer derived by the camphor molecules in two-dimensional space, we need to consider three equations: reaction-diffusion equation for camphor molecules, net force equation, and net torque equation exerted on the swimmer.

$$\frac{\partial \Gamma}{\partial t} = D \nabla^2 \Gamma - k \Gamma + f(x, y) \quad (1)$$

$$(\gamma = \gamma_0 - p \Gamma)$$

The first term on the right-hand side depicted surface diffusion of camphor molecules. The second term means the sublimation (from the water surface to the air) and dissolution (from water surface to the bulk water phase) of the camphor molecules. The last term is the supply of the camphor molecules from the camphor engine, where $f(x, y)$ is described as f_0 in the region in two-dimensional space that corresponds to the shape of the camphor engine, and 0 in the other region. We presume a linear relation between γ and Γ for simplicity. (γ is a surface tension.)

$$\sum F = F_{\text{Driving force 1}} + F_{\text{Driving force 2}} + F_{\text{drag}} \quad (2)$$

$$\sum \tau = \tau_{F_{\text{Driving force 1}} + F_{\text{Driving force 2}}} + \tau_{\text{drag}} \quad (3)$$

Above equations (2) and (3) are the net force equation and net torque equation. We need to consider the force and torque exerted by the camphor molecules and also include drag forces and drag torques. Then we are able to calculate the trajectory of the Marangoni swimmer by solving the above equations.

Previous literature has extensively discussed the modeling of driving forces generated by camphor engines in various systems¹⁻⁷. When neglecting factors contributing to the system's complexity, such as the interaction between camphor engines, the swimmer's shape, and external forces, the driving force of the camphor engine can be modeled in a linear term in one-dimensional system, with a direction vector perpendicular to the edge on which the camphor engines are placed. These assumptions neglect the nonlinear terms, fluid dynamics complexities and consider only the dominant forces. As a result, the simplified linear model may provide an approximate prediction. However, these assumptions can simplify the model and reduce the computational complexity of the prediction, making it a useful tool for analyzing the motion of Marangoni swimmers with camphor engines.

The driving force generated by the camphor engine, as shown below, can be derived from Reaction-diffusion and Newtonian equations (simplified version in one-dimensional space). Based on the previous report (N. J. Suematsu et al., Langmuir 2014)⁸, when solving these two equations, the steady-state swimmers' driving force and velocity have a relationship as expressed in Equation (7).

Reaction–diffusion equation:

$$\frac{\partial \Gamma(x,t)}{\partial t} = D \nabla^2 \Gamma(x,t) - \kappa \Gamma(x,t) + \alpha L^{-1} \delta(x - x_0) \quad (4)$$

where the delta function, δ , denotes the assumption that the camphor molecules are supplied only to the stern of the boat⁹.

Newtonian equation:

$$m \frac{d^2 x_0}{dt^2} = -\mu \frac{dx_0}{dt} + F_w = 0 \quad (5)$$

$$F_w = L(\gamma_0 - \gamma(\Gamma(x_0, t))) \quad (6)$$

$$F_{w0} = \mu v_{steady} \sqrt{\frac{v_{steady}^2}{4kD} + 1} \quad (7)$$

By inputting the calculated driving force into the system, it becomes relatively easy to estimate the object's trajectory. However, due to the difficulties in calculating this approach for every system with varying shapes and designs, we utilized FEA simulation to predict the trajectories of various swimmers.

Commercially available finite element analysis (FEA) software, ABAQUS, was used to predict the trajectories of the swimmers. To derive the driving pressure (driving force per unit area of a camphor engine), we first estimated the approximate value of the friction coefficient (μ) based on the previous research conducted by N. Suematsu and colleagues⁸. We designed an experimental condition that mimics the situation where the drag force becomes dominant (driving force negligible) by using swimmers with not-fully-dried camphor engines, which results in a burst release followed by a weakened release of camphor molecules. The experimental data shows relaxation-like motion after the end of the burst release of camphor molecules from the not-fully-dried camphor engines. At the relaxation interval, the drag force becomes dominant, making the velocity of the Marangoni swimmer decrease exponentially. Then the equation (5) can be reasonably approximated and solved as equation (8), where m is the mass of the boat (0.07 g) and v_{init} is the initial velocity of the swimmer.

$$v(t) = v_{init} e^{-\frac{\mu}{m}t} \quad (8)$$

Then the value of μ ($3.327 \cdot 10^{-5} \text{ Nm}^{-1}\text{s}$) can be derived from the slope of $\ln(v)$. Subsequently, we calculated the driving force F_{w0} of each camphor engine of different concentrations from their steady-state velocity v_{steady} using the following equation (7).

Because the constants k and D are broadly applicable to a variety of situations regardless of the shape of the swimmer, we used the values $k = 2 \cdot 10^{-2} \text{ s}^{-1}$ and $D = 4 \cdot 10^{-3} \text{ m}^2\text{s}^{-1}$ adopted from a previous report (N. J. Suematsu et al., Langmuir 2014)⁸. The driving pressure was then calculated for each concentration by dividing the driving force by the working area (the side area of the swimmer where the camphor ink was drawn, $A = 3 \text{ mm}^2$). For the camphor inks applied in dots, we analyzed the diameter of the dots ($\sim 2.5 \text{ mm}$) and applied the driving pressure to the nearest side of the swimmer with equal length. For the dots located in the vertex, the driving pressure was applied to both nearest sides of the swimmer.

Finally, stagnation pressure P_s was applied to the edges of the swimmer, as computed by the following equation:

$$P_s = -c_s (v \cdot n - v_{\text{ref}} \cdot n)^2 \quad (9)$$

where c_s is the fitted coefficient ($c_s = 1.4 \text{ kg/m}^3$) from the circular trajectory in Fig. 2e, n is the normal unit outward from the element where the surface pressure is applied, and v_{ref} is the velocity of the reference node.

In all simulations, the swimmers with length = 20 mm, width = 10 mm, and height = 0.3 mm were constructed with 3D deformable solid elements of type C3D8 using a linear elastic model. The mechanical properties were determined by using the following input parameters: density $d = 1.45 \text{ g/cm}^3$, Young's modulus $E = 3.275 \text{ GPa}$, and Poisson's ratio $\nu = 0.4$.

Thank you again for your feedback and for considering our manuscript for publication in Nature Communications.

Response to reviewer #2's comments

General Comment: The authors describe a technique for producing Marangoni swimmers and other objects that generate Marangoni flows. The technique is based on loading a marker pen with ink that contains camphor. Using the pen to write on otherwise generic objects patterns them with local stores of camphor that are gradually released and cause Marangoni flows when the surface contacts water. The authors demonstrate this with pieces of PET film, acrylic, paper, leaves, and 3D printed structures. Moreover, the authors demonstrate that such Marangoni swimmers can be designed to move linearly, spin around, or perform more complex manoeuvres, such as following a maze-like confined path by combining a pattern that generates forward propulsion with a pattern that causes side-to-side oscillations.

To add further complexity, the authors used folded paper, which gradually unfolds upon contact with water to expose different patterns to the water. Different conformations of the structure could be associated with different Marangoni-driven motion.

The novelty in this work is the pen-drawing technique of patterning objects for Marangoni flow. This is significant because it allows very quick and convenient production of swimmers with different patterns, which could facilitate further detailed studies.

The authors have shown a good variety of examples from simple to complex, time-dependent behaviours. In my opinion, the work is of good quality and sufficiently impactful to warrant consideration for publication in Nature Communications. The manuscript is clearly written and figures are well made. Experimental procedures appear to be sound and adequately described, but this aspect is beyond my expertise.

Response for General Comment: We are grateful for the reviewer's time and expertise to carefully reviewing our work and for the insightful comment. We sincerely apologize for any misunderstandings or unintentional misleadings that have arisen regarding the theoretical analysis and simulation part. Thanks to the reviewer's comment, we could be aware of the critical flaws that we have stated in the previous manuscript. We appreciate the opportunity to address these concerns and improve our manuscript, ensuring the clarity and accuracy of our work.

Comment 1: The theoretical modeling and simulation results, however, were somewhat unclear and weak. From the descriptions in the main text and supplementary text, the friction coefficient μ and the driving force are inputs for the finite element analysis. It is not clear what the purpose of the FEA is, then. Mechanical properties of the material are quoted, so presumably strains are calculated but they are not discussed at all in the text. It seems that FEA was only used to calculate trajectories of the swimmers but this approach is not necessary if one already has a formula for the driving force. One can directly solve the equations of motion (2) and (4).

Response 1: We appreciate your observation that the purpose of the FEA in our study might have been unclearly described in the previous manuscript. As pointed out, our initial aim was to predict the trajectories of the swimmers using FEA. It is true that trajectories of various swimmers can be calculated using theoretical modeling performed by other researchers, but

such calculations can be relatively complex to be applied in every swimmer with diverse design. Therefore, we attempted to simplify the prediction of the trajectories by employing a simplified linear model (1-dimensional model) to compute the driving force of various concentrations of camphor ink and applying these forces locally to the spot of the swimmer where the engines are drawn (2-dimensional model). We acknowledge that this might not have been clearly conveyed in the previous manuscript, and we have revised it accordingly to better articulate our approach and its rationale. Based on another reviewer's comment, we have moved all the theoretical modeling equations to the Supplementary Information, rather than partially explaining them in the Manuscript.

Manuscript

We show that the motion trajectory of a swimmer can be programmed using a drawing pattern (Fig. 2e and Supplementary Movie 3). We tested three different patterns: two camphor-ink dots on the vehicle but with different arrangements. Depending on the location of the dots (i.e., pattern of camphor-ink drawing), the swimmers have different trajectories, such as linear, circular, or rotational motion. The trajectory of the Marangoni swimmer can be mathematically modeled by calculating the driving force generated due to camphor release and assuming that the force acts on the camphor-ink-drawn area (referred to as the camphor engine hereinafter) (See Supplementary Text)^{30,37–39}. However, due to the complexity of factoring in the swimmer's shape and the camphor engine design, we opted to implement this estimation through the finite element analysis (FEA) simulation (Fig. 2e) (see Supplementary Text).

Supplementary information

Mathematical modeling and FEA simulation for predicting camphor-driven swimmer trajectories

Table. Parameters for mathematical modeling and FEA simulation

Notation	Description	Unit
$\Gamma(x, y, t)$, $\Gamma(x, t)$, $\bar{\Gamma}$	Surface camphor concentration Notation depends on modeling dimensions	$[\text{mol m}^{-2}]$
$f(x, y)$	Supply of the camphor molecules from the camphor engine to the water surface (in 2-dimension modeling)	$[\text{mol s}^{-1}\text{m}^{-1}]$
γ	Surface tension	$[\text{N m}^{-1}]$

γ_0	Surface tension of pure water	$[N\ m^{-1}]$
p	Positive constant	
$F_{\text{Driving force}}$	Driving force of the moving camphor object (in 2-dimensional modeling)	$[N]$
F_{drag}	Drag force exerted on the object (in 2-dimensional modeling)	$[N]$
D	Diffusion constant	$[m^2\ s^{-1}]$
k	Sum of k_s and k_d	$[s^{-1}]$
k_s	Sublimation rate from the water surface	$[s^{-1}]$
k_d	Dissolution rate from the water surface	$[s^{-1}]$
x_0	Position at the edge of the camphor engine	$[m]$
m	Mass of the camphor boat	$[kg]$
μ	Friction coefficient	$[N\ m^{-1}\ s]$
F_w	Driving force of the moving camphor boat (in 1-dimensional modeling)	$[N]$
F_{w0}	Driving force of the stationary camphor boat (in 1-dimensional modeling)	$[N]$
L	Length of the contact line between the camphor disk and water surface	$[m]$

 v_{steady}

Steady state velocity of camphor engine

 $[m s^{-1}]$

 α Supply rate of camphor molecules from the camphor engine to the water surface
(in 1-dimensional modeling) $[mol s^{-1}]$

Predicting an object's trajectory can be achieved through a comprehensive analysis of the net force and net torque applied to the object. By understanding these factors, it becomes possible to anticipate the motion and behavior of the Marangoni swimmer in various situations.

For example, let's consider a rectangular Marangoni swimmer with two camphor engines placed on one edge, as shown below.

We refer to the two-dimensional modeling of a Marangoni swimmer's motion studied by H. Kitahata and colleagues¹. The surface concentration of camphor molecules is defined as $\Gamma(x, y, t)$, which is represented as Γ for simple notation in equation (1). To figure out the trajectories of the swimmer derived by the camphor molecules in two-dimensional space, we need to consider three equations: reaction-diffusion equation for camphor molecules, net force equation, and net torque equation exerted on the swimmer.

$$\frac{\partial \Gamma}{\partial t} = D \nabla^2 \Gamma - k \Gamma + f(x, y) \quad (1)$$
$$(\gamma = \gamma_0 - p \Gamma)$$

The first term on the right-hand side depicted surface diffusion of camphor molecules. The second term means the sublimation (from the water surface to the air) and dissolution (from water surface to the bulk water phase) of the camphor molecules. The last term is the supply of the camphor molecules from the camphor engine, where $f(x, y)$ is described as f_0 in the region in two-dimensional space that corresponds to the shape of the camphor engine, and 0 in the other region. We presume a linear relation between γ and Γ for simplicity. (γ is a surface tension.)

$$\sum F = F_{Driving force 1} + F_{Driving force 2} + F_{drag} \quad (2)$$

$$\sum \tau = \tau_{F_{Driving force 1} + F_{Driving force 2}} + \tau_{drag} \quad (3)$$

Above equations (2) and (3) are the net force equation and net torque equation. We need to consider the force and torque exerted by the camphor molecules and also include drag forces

and drag torques. Then we are able to calculate the trajectory of the Marangoni swimmer by solving the above equations.

Previous literature has extensively discussed the modeling of driving forces generated by camphor engines in various systems¹⁻⁷. When neglecting factors contributing to the system's complexity, such as the interaction between camphor engines, the swimmer's shape, and external forces, the driving force of the camphor engine can be modeled in a linear term in one-dimensional system, with a direction vector perpendicular to the edge on which the camphor engines are placed. These assumptions neglect the nonlinear terms, fluid dynamics complexities and consider only the dominant forces. As a result, the simplified linear model may provide an approximate prediction. However, these assumptions can simplify the model and reduce the computational complexity of the prediction, making it a useful tool for analyzing the motion of Marangoni swimmers with camphor engines.

The driving force generated by the camphor engine, as shown below, can be derived from Reaction-diffusion and Newtonian equations (simplified version in one-dimensional space). Based on the previous report (N. J. Suematsu et al., Langmuir 2014)⁸, when solving these two equations, the steady-state swimmers' driving force and velocity have a relationship as expressed in Equation (7).

Reaction-diffusion equation:

$$\frac{\partial \Gamma(x,t)}{\partial t} = D \nabla^2 \Gamma(x,t) - \kappa \Gamma(x,t) + \alpha L^{-1} \delta(x - x_0) \quad (4)$$

where the delta function, δ , denotes the assumption that the camphor molecules are supplied only to the stern of the boat⁹.

Newtonian equation:

$$m \frac{d^2 x_0}{dt^2} = -\mu \frac{dx_0}{dt} + F_w = 0 \quad (5)$$

$$F_w = L(\gamma_0 - \gamma(\Gamma(x_0, t))) \quad (6)$$

$$F_{w0} = \mu v_{steady} \sqrt{\frac{v_{steady}^2}{4kD} + 1} \quad (7)$$

By inputting the calculated driving force into the system, it becomes relatively easy to estimate the object's trajectory. However, due to the difficulties in calculating this approach for every system with varying shapes and designs, we utilized FEA simulation to predict the trajectories of various swimmers.

Commercially available finite element analysis (FEA) software, ABAQUS, was used to predict the trajectories of the swimmers. To derive the driving pressure (driving force per unit area of a camphor engine), we first estimated the approximate value of the friction coefficient (μ) based on the previous research conducted by N. Suematsu and colleagues⁸. We designed an experimental condition that mimics the situation where the drag force becomes dominant (driving force negligible) by using swimmers with not-fully-dried camphor engines, which results in a burst release followed by a weakened release of camphor molecules. The experimental data shows relaxation-like motion after the end of the burst release of camphor molecules from the not-fully-dried camphor engines. At the relaxation interval, the drag force becomes dominant, making the velocity of the Marangoni swimmer decrease exponentially. Then the equation (5) can be reasonably approximated and solved as equation (8), where m is the mass of the boat (0.07 g) and v_{init} is the initial velocity of the swimmer.

$$v(t) = v_{init} e^{-\frac{\mu}{m}t} \quad (8)$$

Then the value of μ ($3.327 \cdot 10^{-5} \text{ Nm}^{-1}\text{s}$) can be derived from the slope of $\ln(v)$. Subsequently, we calculated the driving force F_{w0} of each camphor engine of different concentrations from their steady-state velocity v_{steady} using the following equation (7).

Because the constants k and D are broadly applicable to a variety of situations regardless of the shape of the swimmer, we used the values $k = 2 \cdot 10^{-2} \text{ s}^{-1}$ and $D = 4 \cdot 10^{-3} \text{ m}^2\text{s}^{-1}$ adopted from a previous report (N. J. Suematsu et al., Langmuir 2014)⁸. The driving pressure was then calculated for each concentration by dividing the driving force by the working area (the side area of the swimmer where the camphor ink was drawn, $A = 3 \text{ mm}^2$). For the camphor inks applied in dots, we analyzed the diameter of the dots ($\sim 2.5 \text{ mm}$) and applied the driving pressure to the nearest side of the swimmer with equal length. For the dots located in the vertex, the driving pressure was applied to both nearest sides of the swimmer.

Finally, stagnation pressure P_s was applied to the edges of the swimmer, as computed by the following equation:

$$P_s = -c_s (v \cdot n - v_{ref} \cdot n)^2 \quad (9)$$

where c_s is the fitted coefficient ($c_s = 1.4 \text{ kg/m}^3$) from the circular trajectory in Fig. 2e, n is the normal unit outward from the element where the surface pressure is applied, and v_{ref} is the velocity of the reference node.

In all simulations, the swimmers with length = 20 mm, width = 10 mm, and height = 0.3 mm were constructed with 3D deformable solid elements of type C3D8 using a linear elastic model. The mechanical properties were determined by using the following input parameters: density $d = 1.45 \text{ g/cm}^3$, Young's modulus $E = 3.275 \text{ GPa}$, and Poisson's ratio $\nu = 0.4$.

We hope that these revisions address your concerns and appreciate your guidance in improving our manuscript.

Comment 2: The principles that are demonstrated in this work are interesting but there is scope for better quantitative modeling of the motions produced by different patterns. The theoretical analysis in the supplementary text attempts to address this but appears to have some significant flaws. For instance, it is assumed that the direction of the forces at the patterned corners is vertical (as shown in the supplementary figure) but no explanation is given for this. Moreover, in the process of deriving the magnitude of these forces, it is stated that (i) the spatial gradient of camphor is negligible and that (ii) the concentration of released camphor molecules is negligible. Each of these assumptions should mean that the surface tension is the same everywhere, i.e., there is no Marangoni flow and no motion of the swimmer. Even with the presentation clarified, the basis of this model is the formula for the force proposed in Ref 27, which was based on linear motion and therefore had an unambiguous direction even if details of the (symmetric) shape were omitted. For rotational motion, the direction of the forces each ink patch produces must depend on the shape of the swimmer.

Response 2: We acknowledge that there were significant mistakes in our model, and we apologize for any confusion this may have caused. We appreciate the opportunity to correct these errors and ensure the accuracy of our work. Thanks to the reviewer's feedback, we have become aware of the critical flaws in theoretical analysis, which we had previously misunderstood. Reviewer's guidance has been invaluable in helping us identify the areas that require revision and improvement. As a result, we have revisited and revised the entire theoretical analysis section to address the issues the reviewer raised, as described in **Response 1**.

To address the particular issues that you have raised in Comment 2, please find our detailed responses below:

- 1) For instance, it is assumed that the direction of the forces at the patterned corners is vertical (as shown in the supplementary figure) but no explanation is given for this.

In response to the comment regarding the assumption that the direction of the forces

at the patterned corners is vertical, we apologize for any confusion our initial explanation may have caused. We changed the example to explain the trajectory of the swimmer as shown below

For example, let's consider a rectangular Marangoni swimmer with two camphor engines placed on one edge, as shown below.

$$\Sigma F = F_{\text{Driving force 1}} + F_{\text{Driving force 2}} + F_{\text{drag}} \quad (2)$$

$$\Sigma \tau = \tau_{F_{\text{Driving force 1}} + F_{\text{Driving force 2}}} + \tau_{\text{drag}} \quad (3)$$

Above equations (2) and (3) are the net force equation and net torque equation. We need to consider the force and torque exerted by the camphor molecules and also include drag forces and drag torques. Then we are able to calculate the trajectory of the Marangoni swimmer by solving the above equations.

Additionally, we would like to emphasize that both vertical and horizontal forces are included in our FEA simulations. We have revised the simulation section of our manuscript to highlight this aspect. We believe that this revised approach provides a more accurate representation of the forces at the patterned corners and is better suited for our system. Thank you for bringing this important matter to our attention.

For the camphor inks applied in dots, we analyzed the diameter of the dots (~ 2.5 mm) and applied the driving pressure to the nearest side of the swimmer with equal length. For the dots located in the vertex, the driving pressure was applied to both nearest sides of the swimmer.

- 2) Moreover, in the process of deriving the magnitude of these forces, it is stated that (i) the spatial gradient of camphor is negligible and that (ii) the concentration of released camphor molecules is negligible. Each of these assumptions should mean that the surface tension is the same everywhere, i.e., there is no Marangoni flow and no motion of the swimmer.

We apologize for the oversight and have since revised our theoretical modeling to address these issues. The updated modeling now provides a more accurate representation of the Marangoni flow and the motion of the swimmer. Please find the

revised theoretical modeling and corresponding discussions in the updated manuscript.

The driving force generated by the camphor engine, as shown below, can be derived from Reaction-diffusion and Newtonian equations (simplified version in one-dimensional space). Based on the previous report (N. J. Suematsu et al., Langmuir 2014)⁸, when solving these two equations, the steady-state swimmers' driving force and velocity have a relationship as expressed in Equation (7).

Reaction-diffusion equation:

$$\frac{\partial \Gamma(x,t)}{\partial t} = D \nabla^2 \Gamma(x,t) - \kappa \Gamma(x,t) + \alpha L^{-1} \delta(x - x_0) \quad (4)$$

where the delta function, δ , denotes the assumption that the camphor molecules are supplied only to the stern of the boat⁹.

Newtonian equation:

$$m \frac{d^2 x_0}{dt^2} = -\mu \frac{dx_0}{dt} + F_w = 0 \quad (5)$$

$$F_w = L(\gamma_0 - \gamma(\Gamma(x_0, t))) \quad (6)$$

$$F_{w0} = \mu v_{steady} \sqrt{\frac{v_{steady}^2}{4kD} + 1} \quad (7)$$

We are grateful for your valuable feedback, which has helped us identify and correct this crucial error, and we hope that our revised modeling now adequately addresses your concerns.

- 3) Even with the presentation clarified, the basis of this model is the formula for the force proposed in Ref 27, which was based on linear motion and therefore had an unambiguous direction even if details of the (symmetric) shape were omitted. For rotational motion, the direction of the forces each ink patch produces must depend on the shape of the swimmer.

We understand your concern about the applicability of this formula to rotational motion, given that it was originally derived for linear motion with an unambiguous direction. In our study, we utilized the formula for linear motion to calculate the

driving force in a simplified manner, and assumed that the force is acting locally on the camphor engine using FEA simulation. We acknowledge that this approach may not fully capture the complex dynamics required to precisely estimate the trajectory of the swimmer. However, it does allow us to easily predict the swimmer's trajectory while considering the swimmer's shape, without the need to solve complex equations.

We have included a detailed explanation of our methodology, as well as the limitations of our current approach, in both the manuscript and the supplementary information. Additionally, we have referenced other research that has more accurately calculated the motion of similar systems. This context helps to highlight the trade-offs made in our study for the sake of simplicity, and provides a foundation for potential improvements in future work.

Previous literature has extensively discussed the modeling of driving forces generated by camphor engines in various systems¹⁻⁷. When neglecting factors contributing to the system's complexity, such as the interaction between camphor engines, the swimmer's shape, and external forces, the driving force of the camphor engine can be modeled in a linear term in one-dimensional system, with a direction vector perpendicular to the edge on which the camphor engines are placed. These assumptions neglect the nonlinear terms, fluid dynamics complexities and consider only the dominant forces. As a result, the simplified linear model may provide an approximate prediction. However, these assumptions can simplify the model and reduce the computational complexity of the prediction, making it a useful tool for analyzing the motion of Marangoni swimmers with camphor engines.

We believe that these changes will significantly enhance the quality and rigor of our work, providing a more comprehensive understanding of the principles governing the motion of Marangoni swimmers with different patterns.

Comment 3: In equation 2 of the main text and equation 3 of the supplementary material, r and r_0 should be γ and γ_0 , respectively. Also, in the supplementary text, equations 6-7 have missed the parentheses around $\Gamma(x,t)$, making it look like a product of γ and Γ whereas actually Γ is the argument of γ .

Response 3: We would like to extend our sincere apologies for any typos present in the manuscript. We understand that such errors can make it difficult to read and comprehend the content, especially in the equation part, and we deeply regret any inconvenience this may have caused during the review process. We carefully proofread the manuscript and corrected all typos. We appreciate the reviewer for the understanding and thank once again for the valuable feedback.

Comment 4: "The mean surface concentration can be derived using equation (3)... This

implies that Marangoni propulsion, which is the difference in the surface tension around the vehicle, increases when the release ratio of the camphor molecules increases." It is reasonable that the propulsion increases when the release ratio increases but this is not related to equation (3), which only describes the mean concentration (which does not contribute to Marangoni propulsion).

Response 4: We would like to reiterate our appreciation for your insightful feedback, which has been crucial in helping us identify and address the critical flaws in our theoretical analysis. Your guidance has played a pivotal role in our understanding of the issues at hand, allowing us to significantly improve our work. By revising the entire theoretical analysis section, we have been able to correct our previous misunderstandings and ensure the accuracy of our research. We would like to kindly inform the reviewer that we have carefully addressed the concerns raised in the following manner:

The driving force generated by the camphor engine, as shown below, can be derived from Reaction-diffusion and Newtonian equations (simplified version in one-dimensional space). Based on the previous report (N. J. Suematsu et al., Langmuir 2014)⁸, when solving these two equations, the steady-state swimmers' driving force and velocity have a relationship as expressed in Equation (7).

Reaction-diffusion equation:

$$\frac{\partial \Gamma(x,t)}{\partial t} = D \nabla^2 \Gamma(x,t) - \kappa \Gamma(x,t) + \alpha L^{-1} \delta(x - x_0) \quad (4)$$

where the delta function, δ , denotes the assumption that the camphor molecules are supplied only to the stern of the boat⁹.

Newtonian equation:

$$m \frac{d^2 x_0}{dt^2} = -\mu \frac{dx_0}{dt} + F_w = 0 \quad (5)$$

$$F_w = L(\gamma_0 - \gamma(\Gamma(x_0, t))) \quad (6)$$

$$F_{w0} = \mu v_{steady} \sqrt{\frac{v_{steady}^2}{4kD} + 1} \quad (7)$$

We are truly grateful for your valuable input and believe that, thanks to your comments, our manuscript has been greatly strengthened.

Comment 5: Variables in the equations should be defined. Also, the mean surface concentration in equation (3) should be distinguished from the spatially-dependent variable, e.g., with a bar above the Gamma symbol.

Response 5: We sincerely apologize for misleadings that the previous equations have caused. We have thoroughly revisited the theoretical part of our study and have revised the issues raised in comment 5. Also, we defined all the variables in the theoretical remodeling part as follows.

Table. Parameters for mathematical modeling and FEA simulation

Notation	Description	Unit
$\Gamma(x, y, t),$ $\bar{\Gamma}(x, t), \bar{\Gamma}$	Surface camphor concentration Notation depends on modeling dimensions	$[\text{mol m}^{-2}]$
$f(x, y)$	Supply of the camphor molecules from the camphor engine to the water surface (in 2-dimension modeling)	$[\text{mol s}^{-1}\text{m}^{-1}]$
γ	Surface tension	$[\text{N m}^{-1}]$
γ_0	Surface tension of pure water	$[\text{N m}^{-1}]$
p	Positive constant	
$F_{\text{Driving force}}$	Driving force of the moving camphor object (in 2-dimensional modeling)	$[\text{N}]$
F_{drag}	Drag force exerted on the object (in 2-dimensional modeling)	$[\text{N}]$
D	Diffusion constant	$[\text{m}^2 \text{s}^{-1}]$

k	Sum of k_s and k_d	$[s^{-1}]$
k_s	Sublimation rate from the water surface	$[s^{-1}]$
k_d	Dissolution rate from the water surface	$[s^{-1}]$
x_0	Position at the edge of the camphor engine	$[m]$
m	Mass of the camphor boat	$[kg]$
μ	Friction coefficient	$[N\ m^{-1}\ s]$
F_w	Driving force of the moving camphor boat (in 1-dimensional modeling)	$[N]$
F_{w0}	Driving force of the stationary camphor boat (in 1-dimensional modeling)	$[N]$
L	Length of the contact line between the camphor disk and water surface	$[m]$
v_{steady}	Steady state velocity of camphor engine	$[m\ s^{-1}]$
α	Supply rate of camphor molecules from the camphor engine to the water surface (in 1-dimensional modeling)	$[mol\ s^{-1}]$

Comment 6: Details of the FEA were missing, such as shape and dimensions of the 3D object. What direction is the driving pressure applied for the dots? Following figures in the manuscript, the dots should be placed on the horizontal surfaces so pressure would act vertically and not drive horizontal motion.

Response 6: We appreciate your attention to the details of the finite element analysis (FEA) and concern regarding the shape, dimensions, and driving pressure direction of the 3D object.

We understand that providing these details is important for a clear understanding of our methodology. In response to this comment, we have added the necessary information regarding the simulation in Supplementary Information as follows.

Commercially available finite element analysis (FEA) software, ABAQUS, was used to predict the trajectories of the swimmers. To derive the driving pressure (driving force per unit area of a camphor engine), we first estimated the approximate value of the friction coefficient (μ) based on the previous research conducted by N. Suematsu and colleagues⁸. We designed an experimental condition that mimics the situation where the drag force becomes dominant (driving force negligible) by using swimmers with not-fully-dried camphor engines, which results in a burst release followed by a weakened release of camphor molecules. The experimental data shows relaxation-like motion after the end of the burst release of camphor molecules from the not-fully-dried camphor engines. At the relaxation interval, the drag force becomes dominant, making the velocity of the Marangoni swimmer decrease exponentially. Then the equation (5) can be reasonably approximated and solved as equation (8), where m is the mass of the boat (0.07 g) and v_{init} is the initial velocity of the swimmer.

$$v(t) = v_{init} e^{-\frac{\mu}{m}t} \quad (8)$$

Then the value of μ ($3.327 \cdot 10^{-5} \text{ Nm}^{-1}\text{s}$) can be derived from the slope of $\ln(v)$. Subsequently, we calculated the driving force F_{w0} of each camphor engine of different concentrations from their steady-state velocity v_{steady} using the following equation (7).

Because the constants k and D are broadly applicable to a variety of situations regardless of the shape of the swimmer, we used the values $k = 2 \cdot 10^{-2} \text{ s}^{-1}$ and $D = 4 \cdot 10^{-3} \text{ m}^2\text{s}^{-1}$ adopted from a previous report (N. J. Suematsu et al., Langmuir 2014)⁸. The driving pressure was then calculated for each concentration by dividing the driving force by the working area (the side area of the swimmer where the camphor ink was drawn, $A = 3 \text{ mm}^2$). For the camphor inks applied in dots, we analyzed the diameter of the dots ($\sim 2.5 \text{ mm}$) and applied the driving pressure to the nearest side of the swimmer with equal length. For the dots located in the vertex, the driving pressure was applied to both nearest sides of the swimmer.

Finally, stagnation pressure P_s was applied to the edges of the swimmer, as computed by the following equation:

$$P_s = -c_s (v \cdot n - v_{ref} \cdot n)^2 \quad (9)$$

where c_s is the fitted coefficient ($c_s = 1.4 \text{ kg/m}^3$) from the circular trajectory in Fig. 2e, n is the normal unit outward from the element where the surface pressure is applied, and v_{ref} is the velocity of the reference node.

In all simulations, the swimmers with length = 20 mm, width = 10 mm, and height = 0.3 mm were constructed with 3D deformable solid elements of type C3D8 using a linear elastic model. The mechanical properties were determined by using the following input parameters: density $d = 1.45 \text{ g/cm}^3$, Young's modulus $E = 3.275 \text{ GPa}$, and Poisson's ratio $\nu = 0.4$.

Comment 7: "We demonstrated a square-shaped trajectory by selecting the optimal concentration of the camphor ink (Fig 3b, right)." This suggests that the camphor concentration is the main parameter that needs to be tuned but it must surely also be somewhat sensitive to how the pattern is drawn, e.g., thickness and lengths of lines, radius of dots. More generally, could the authors comment on how reproducible the behavior of swimmers was? Is it possible to reliably make swimmers with the same motion? Closed, periodic trajectories such as the square and star-shaped paths should be especially difficult to control accurately.

Response 7: Thank you for your comment regarding the reproducibility of our work. We have added some more experimental data (Supplementary Fig. 3) to demonstrate the reproducibility of complex motion programming and believe that this will further improve the quality of this manuscript.

In our first demonstration of complex motion programming, the star-shaped trajectory, we observed fairly consistent moving modalities across multiple trials. The swimmers consistently traveled along the boundary while bouncing off walls. Out of five additional attempts, all succeeded in obtaining a similar moving trajectory. However, the angle between vertices, influenced by the balance between oscillation moving speed and rotation speed, varied among trials. Two out of the five trials resulted in a 4-pointed star-shaped trajectory. We acknowledge that this variability may stem from the intrinsic sensitivity of manual hand drawing (e.g., thickness and lengths of lines, radius of dots), as the reviewer have pointed out. We anticipate that the accuracy can be improved if we adopt a more accurate fabrication method, such as ink-jet based printing or lithography.

Our second demonstration, the polygonal trajectory, also displayed high reproducibility, with all five attempts resulting in successful trials. However, similar to the star-shaped trajectory, the rotation angle at each vertex varied among trials, leading one of the five attempts to produce a hexagonal trajectory instead of a square trajectory. We attribute this variation to the inaccuracy caused by manual hand drawing. To better convey our findings, we have opted to use the term "polygonal trajectory" rather than "square trajectory."

To enhance the clarity of our manuscript, we have included additional explanations and experimental data in the main text and Supplementary materials, addressing the concerns you raised.

Manuscript

We investigated the reproducibility of these complex motion programming, as shown in **Supplementary Fig. 3**. First, for the programming of the star-shaped trajectory, all five attempts from the five trials were successful in drawing star-shaped trajectories that traveled along the boundary while bouncing off a wall (**Supplementary Fig. 3a**). However, there were some differences in the angle between each vertex depending on the attempt, which seems to be due to the inaccuracy caused by manual hand drawing. Three out of five attempts showed a five-pointed star shape, and the other two had approximately a four-pointed star shape. Secondly, the reproducibility of the polygonal trajectory was investigated as shown in **Supplementary Fig. 3b**. Every experiment from five attempts showed a polygonal trajectory with periodic movement, but one of them had a lack of rotation angle in the vertex, which led to a hexagonal path, rather than a square path like the others. We believe that this inaccuracy can be improved if we adopt a more precise printing method rather than manual hand drawing.

Supplementary Information

Supplementary Fig. 3. Reproducibility of complex motion programming. **a** Reproducibility of star-shaped trajectory programming. Three out of five attempts had about five vertices during one round of the boundary of the water tank. The other two trials showed about four vertices per one round. **b** Reproducibility of polygonal trajectory programming. Four out of five attempts showed square like trajectories while the other one trial showed hexagonal trajectory.

Comment 8: The authors mention that the dissolution of the water-soluble bridge causes a boost in motion as the bonded swimmers separate. How does the film affect motion before separation? Can the film itself be used as a Marangoni fuel? Were any control cases tested for initially bonded swimmers where no camphor ink is used?

Response 8: Thank you for your thoughtful comment and inquiry into the potential effects of the water-soluble bridge on the Marangoni fuel propulsion. We appreciate the opportunity to provide further clarification on this matter.

We acknowledge that the pullulan film could potentially act as a Marangoni fuel by reducing the surface tension of water as it dissolves. To minimize any undesired effects on the swimmer's trajectory, we carefully designed the bridging films with the following considerations: Firstly, the pullulan films were designed to be bilaterally symmetrical around the direction of progress before disassembly. Thus, the pullulan film does not exert net force in the direction vertical to the heading direction. Secondly, the dissolvable area is not exposed to the front or back parts of the vehicle, minimizing any forward or backward propulsion force generated by the pullulan film before disassembly. The dissolved pullulan molecules primarily diffuse into the bulk water phase underneath the vehicle (Fig. 7a) or underneath the vehicle and symmetrically to the side (Fig. 7b), which has little effect on the swimmer's movement. In fact, the modeling of Marangoni swimmer movement often overlooks the dissolution of molecules into the bulk water phase (N. J. Suematsu et al., *Langmuir* 2014). Additionally, considering the swimmer's forward movement shown in Fig. 7 before disassembly, we can identify that the bridging film does not generate any unwanted backward propulsion force, which can overcome the propulsion force generated by the camphor engine.

We agree that incorporating this explanation into the manuscript will enhance readers' understanding of the study. Accordingly, we have updated the manuscript (Highlighted in yellow) to include this information.

After the subsidence of the instant boost in motion due to the water-soluble film, the individual swimmers followed the designated locomotion according to their camphor engine design and vehicle shape. Since the bridging film can decrease the surface tension of water as it dissolves and potentially acts as a Marangoni fuel, the location of the bridging films needs to be carefully designed to avoid any unwanted directional changes before disassembly. To achieve this, we designed the pullulan films to be bilaterally symmetrical around the heading direction and to not be exposed to the front or back of the vehicle. This ensures that no rotational, forward or backward propulsion force is generated by the bridging film before disassembly.

Response to reviewer #3's comments

General Comment: The paper reports on a new experimental system on "Marangoni swimmer". A material that releases surface active chemicals to the surrounding water surface is known to exhibit self propulsion. Camphor is one of the most famous chemicals for such motion. When a camphor particle is put on the water surface, it moves spontaneously since surface tension around the particle decreases due to the camphor molecules and the surface tension working on the particle becomes unbalanced. The authors newly designed the material that shows the self-propulsion. The advantage of the material is the ease in molding. The authors claim that they can make the swimmer like "pen-writing". Then, they have shown many examples on the interesting behaviors using the new material. The development of the material is novel and important, and therefore I think the manuscript has a potential to be published in Nature Communications. However, each result shown in the figures and movies has been reported using the other systems. Thus, the authors have to comment on the previous related results. Moreover, the theoretical and numerical parts are poorly written. The manuscript should be reviewed again after the author revises based on the comments.

Response for General Comment: We appreciate the acknowledgement of the novelty and importance of our pen-drawn Marangoni swimmers. We thoroughly reviewed the relevant literature and updated all referred articles that the reviewer has suggested. We also apologize for any shortcomings in the presentation of the theoretical and numerical parts of the manuscript. We are grateful for the opportunity to address these concerns and improve our manuscript.

Comment 1: The theoretical parts are quite unclear. The authors briefly show the equations 1 to 3. However, they do not clearly show what are the variables and what are the constants. Especially, in equations 1 and 2, $\Gamma(x,t)$ looks like a variable, but in equation 3 $\Gamma(t)$ is given. It makes the readers confused.

Response 1: We apologize for the confusion caused by the presentation of the theoretical parts and the lack of clarity regarding the variables and constants in the equations. We understand that our previous explanation may have been unclear and misleading. We have thoroughly revisited the theoretical analysis part, and revised the manuscript and supplementary information to provide a comprehensive explanation of the Marangoni propulsion. In our study, we employed a formula for linear motion to calculate the driving force in a simplified manner, and assumed that the force acts locally at the spot where the engine is drawn to predict the trajectory of the swimmer in 2-dimensional space using FEA simulation. Based on another reviewer's comment, we have moved all the theoretical modeling equations to the Supplementary Information, rather than partially explaining them in the Manuscript. Followings are the modified parts of Manuscript and Supplementary Information regarding the theoretical parts and the simulation part.

Manuscript

We show that the motion trajectory of a swimmer can be programmed using a drawing pattern

(Fig. 2e and Supplementary Movie 3). We tested three different patterns: two camphor-ink dots on the vehicle but with different arrangements. Depending on the location of the dots (i.e., pattern of camphor-ink drawing), the swimmers have different trajectories, such as linear, circular, or rotational motion. The trajectory of the Marangoni swimmer can be mathematically modeled by calculating the driving force generated due to camphor release and assuming that the force acts on the camphor-ink-drawn area (referred to as the camphor engine hereinafter) (See Supplementary Text)^{30,37–39}. However, due to the complexity of factoring in the swimmer's shape and the camphor engine design, we opted to implement this estimation through the finite element analysis (FEA) simulation (Fig. 2e) (see Supplementary Text).

Supplementary information

Mathematical modeling and FEA simulation for predicting camphor-driven swimmer trajectories

Table. Parameters for mathematical modeling and FEA simulation

Notation	Description	Unit
$\Gamma(x, y, t)$, $\Gamma(x, t)$, Γ	Surface camphor concentration Notation depends on modeling dimensions	[mol m ⁻²]
$f(x, y)$	Supply of the camphor molecules from the camphor engine to the water surface (in 2-dimension modeling)	[mol s ⁻¹ m ⁻¹]
γ	Surface tension	[N m ⁻¹]
γ_0	Surface tension of pure water	[N m ⁻¹]
p	Positive constant	
$F_{\text{Driving force}}$	Driving force of the moving camphor object (in 2-dimensional modeling)	[N]
F_{drag}	Drag force exerted on the object (in 2-dimensional modeling)	[N]

D	Diffusion constant	$[m^2 s^{-1}]$
k	Sum of k_s and k_d	$[s^{-1}]$
k_s	Sublimation rate from the water surface	$[s^{-1}]$
k_d	Dissolution rate from the water surface	$[s^{-1}]$
x_0	Position at the edge of the camphor engine	$[m]$
m	Mass of the camphor boat	$[kg]$
μ	Friction coefficient	$[N m^{-1} s]$
F_w	Driving force of the moving camphor boat (in 1-dimensional modeling)	$[N]$
F_{w0}	Driving force of the stationary camphor boat (in 1-dimensional modeling)	$[N]$
L	Length of the contact line between the camphor disk and water surface	$[m]$
v_{steady}	Steady state velocity of camphor engine	$[m s^{-1}]$
α	Supply rate of camphor molecules from the camphor engine to the water surface (in 1-dimensional modeling)	$[mol s^{-1}]$

Predicting an object's trajectory can be achieved through a comprehensive analysis of the net force and net torque applied to the object. By understanding these factors, it becomes possible to anticipate the motion and behavior of the Marangoni swimmer in various situations.

For example, let's consider a rectangular Marangoni swimmer with two camphor engines placed on one edge, as shown below.

We refer to the two-dimensional modeling of a Marangoni swimmer's motion studied by H. Kitahata and colleagues ¹. The surface concentration of camphor molecules is defined as $\Gamma(x, y, t)$, which is represented as Γ for simple notation in equation (1). To figure out the trajectories of the swimmer derived by the camphor molecules in two-dimensional space, we need to consider three equations: reaction-diffusion equation for camphor molecules, net force equation, and net torque equation exerted on the swimmer.

$$\frac{\partial \Gamma}{\partial t} = D \nabla^2 \Gamma - k \Gamma + f(x, y) \quad (1)$$

$$(\gamma = \gamma_0 - p \Gamma)$$

The first term on the right-hand side depicted surface diffusion of camphor molecules. The second term means the sublimation (from the water surface to the air) and dissolution (from water surface to the bulk water phase) of the camphor molecules. The last term is the supply of the camphor molecules from the camphor engine, where $f(x, y)$ is described as f_0 in the region in two-dimensional space that corresponds to the shape of the camphor engine, and 0 in the other region. We presume a linear relation between γ and Γ for simplicity. (γ is a surface tension.)

$$\sum F = F_{\text{Driving force 1}} + F_{\text{Driving force 2}} + F_{\text{drag}} \quad (2)$$

$$\sum \tau = \tau_{F_{\text{Driving force 1}} + F_{\text{Driving force 2}}} + \tau_{\text{drag}} \quad (3)$$

Above equations (2) and (3) are the net force equation and net torque equation. We need to consider the force and torque exerted by the camphor molecules and also include drag forces and drag torques. Then we are able to calculate the trajectory of the Marangoni swimmer by solving the above equations.

Previous literature has extensively discussed the modeling of driving forces generated by camphor engines in various systems ¹⁻⁷. When neglecting factors contributing to the system's complexity, such as the interaction between camphor engines, the swimmer's shape, and external forces, the driving force of the camphor engine can be modeled in a linear term in one-dimensional system, with a direction vector perpendicular to the edge on which the camphor engines are placed. These assumptions neglect the nonlinear terms, fluid dynamics complexities and consider only the dominant forces. As a result, the simplified linear model may provide an approximate prediction. However, these assumptions can simplify the model and reduce the computational complexity of the prediction, making it a useful tool for analyzing the motion of Marangoni swimmers with camphor engines.

The driving force generated by the camphor engine, as shown below, can be derived from Reaction-diffusion and Newtonian equations (simplified version in one-dimensional space). Based on the previous report (N. J. Suematsu et al., Langmuir 2014)⁸, when solving these two equations, the steady-state swimmers' driving force and velocity have a relationship as expressed in Equation (7).

Reaction-diffusion equation:

$$\frac{\partial \Gamma(x,t)}{\partial t} = D \nabla^2 \Gamma(x,t) - \kappa \Gamma(x,t) + \alpha L^{-1} \delta(x - x_0) \quad (4)$$

where the delta function, δ , denotes the assumption that the camphor molecules are supplied only to the stern of the boat⁹.

Newtonian equation:

$$m \frac{d^2 x_0}{dt^2} = -\mu \frac{dx_0}{dt} + F_w = 0 \quad (5)$$

$$F_w = L(\gamma_0 - \gamma(\Gamma(x_0, t))) \quad (6)$$

$$F_{w0} = \mu v_{steady} \sqrt{\frac{v_{steady}^2}{4kD} + 1} \quad (7)$$

By inputting the calculated driving force into the system, it becomes relatively easy to estimate the object's trajectory. However, due to the difficulties in calculating this approach for every system with varying shapes and designs, we utilized FEA simulation to predict the trajectories of various swimmers.

Commercially available finite element analysis (FEA) software, ABAQUS, was used to predict the trajectories of the swimmers. To derive the driving pressure (driving force per unit area of a camphor engine), we first estimated the approximate value of the friction coefficient (μ) based on the previous research conducted by N. Suematsu and colleagues⁸. We designed an experimental condition that mimics the situation where the drag force becomes dominant (driving force negligible) by using swimmers with not-fully-dried camphor engines, which results in a burst release followed by a weakened release of camphor molecules. The

experimental data shows relaxation-like motion after the end of the burst release of camphor molecules from the not-fully-dried camphor engines. At the relaxation interval, the drag force becomes dominant, making the velocity of the Marangoni swimmer decrease exponentially. Then the equation (5) can be reasonably approximated and solved as equation (8), where m is the mass of the boat (0.07 g) and v_{init} is the initial velocity of the swimmer.

$$v(t) = v_{init} e^{-\frac{\mu}{m}t} \quad (8)$$

Then the value of μ ($3.327 \cdot 10^{-5} \text{ Nm}^{-1}\text{s}$) can be derived from the slope of $\ln(v)$. Subsequently, we calculated the driving force F_{w0} of each camphor engine of different concentrations from their steady-state velocity v_{steady} using the following equation (7).

Because the constants k and D are broadly applicable to a variety of situations regardless of the shape of the swimmer, we used the values $k = 2 \cdot 10^{-2} \text{ s}^{-1}$ and $D = 4 \cdot 10^{-3} \text{ m}^2\text{s}^{-1}$ adopted from a previous report (N. J. Suematsu et al., Langmuir 2014)⁸. The driving pressure was then calculated for each concentration by dividing the driving force by the working area (the side area of the swimmer where the camphor ink was drawn, $A = 3 \text{ mm}^2$). For the camphor inks applied in dots, we analyzed the diameter of the dots ($\sim 2.5 \text{ mm}$) and applied the driving pressure to the nearest side of the swimmer with equal length. For the dots located in the vertex, the driving pressure was applied to both nearest sides of the swimmer.

Finally, stagnation pressure P_s was applied to the edges of the swimmer, as computed by the following equation:

$$P_s = -c_s (v \cdot n - v_{ref} \cdot n)^2 \quad (9)$$

where c_s is the fitted coefficient ($c_s = 1.4 \text{ kg/m}^3$) from the circular trajectory in Fig. 2e, n is the normal unit outward from the element where the surface pressure is applied, and v_{ref} is the velocity of the reference node.

In all simulations, the swimmers with length = 20 mm, width = 10 mm, and height = 0.3 mm were constructed with 3D deformable solid elements of type C3D8 using a linear elastic model. The mechanical properties were determined by using the following input parameters: density $d = 1.45 \text{ g/cm}^3$, Young's modulus $E = 3.275 \text{ GPa}$, and Poisson's ratio $\nu = 0.4$.

We hope that these revisions address the your concerns and appreciate the guidance in

improving our manuscript.

Comment 2: As for the theoretical work of camphor particles with various shapes has been reported. The author did not cite the previous work on the theoretical framework. The followings are some of them:

- Mathematical modeling and analysis

M. Nagayama et al., *Physica D* 194, 151 (2004).

- Mathematical modeling and analysis for two-dimensional system

H. Kitahata et al., *Phys. Rev. E* 87, 010901 (2013).

- Review for the camphor particle motion

S. Nakata et al., *Phys. Chem. Chem. Phys.* 17, 10326 (2015).

Response 2: We appreciate the reviewer for pointing out the omission of previous theoretical works on camphor particles with various shapes. We understand the importance of acknowledging and building upon the existing literature in our research. In response to this comment, we have carefully reviewed the provided references, as well as conducted additional literature searches to identify other relevant works. We have now cited and discussed these previous theoretical studies in the revised manuscript to give appropriate credit and to contextualize our work within the broader research landscape. We believe that incorporating these references and discussions will strengthen the theoretical framework of our study and provide a more comprehensive understanding of the subject matter. We apologize for any oversight on our part and are grateful for the reviewer's guidance in helping us to improve our manuscript. Followings are the parts of the Manuscript that cite abovementioned references.

*The trajectory of the Marangoni swimmer can be mathematically modeled by calculating the driving force generated due to camphor release and assuming that the force acts on the camphor-ink-drawn area (referred to as the camphor engine hereinafter) (See **Supplementary Text**)^{30,37-39}.*

Comment 3 and 4: In the supporting information, the authors show the mathematical model for the two-dimensional motion. I cannot totally understand the model in SI. Since the authors consider the two-dimensional motion, I think $\Gamma(x,t)$ should be $\Gamma(x,y,t)$. Thus, the expression $\Gamma(x,t)$ in equation (3) in the main text should no longer work. Related to the comment 3, the force originating from the surface tension should work the whole object. I do not think this fact is not included in the model.

Response 3 and 4: We appreciate your comments and understand that our mathematical model for the two-dimensional motion may have been unclear. Initially, our approach was to calculate

the driving force from the simplified linear equations (1-dimensional model), and apply the driving force to the local spots where engines are drawn using the FEA simulation (2-dimensional simulation). We tried to show one example of how a trajectory can be calculated using a theoretical model in the supporting information. However, in our previous manuscript, we inadvertently used one-dimensional equations of motion to describe the trajectory of a two-dimensional object, which led to confusion.

We acknowledge this critical mistake and greatly apologize for any confusion that our previous equation has made. We have now clarified the equations to calculate the trajectory of the swimmer and have made the necessary corrections in the manuscript and supplementary information. The detailed corrections are as described in **Response 1**.

Comment 5: The authors fail to cite the references that are much related to the present contents:

- Translational and Rotational motion of a camphor particle

S. Nakata et al., *Langmuir* 13, 4454–4458 (1997).

- Design of the self-propelled objects using camphor particles

H. Morohashi et al., *Chem. Phys. Lett.* 721, 104 (2019)

- New material for self-propulsion including polymers

R. J. G. Löffler et al., *Molecules* 26, 3116 (2021).

- Interesting trajectory of self-propulsion

H. Kitahata et al., *Phys. Chem. Chem. Phys.* 24, 20326-20335 (2022).

S. Tanaka et al., *Phys. Rev. E* 91, 032406 (2015).

Response 5: Thank you for your comment regarding the references. We apologize for not including the reference in the previous manuscript. We have now carefully reviewed the literatures and updated our manuscript to include the references as highlighted in yellow. We appreciate your feedback and believe that the updated references will enhance the quality of our work. Followings are the parts of the manuscript that we have included abovementioned references.

Conventional Marangoni swimmers with simple designs have demonstrated limited motion capabilities such as linear or circular motion ¹³⁻¹⁷.

The Marangoni swimmer is an aquatic robot that actuates on a liquid surface by releasing surface tension-lowering molecules around its body ¹¹.

Several studies have investigated swimmers with similar trajectories ^{40,41}; *however, to the best of our knowledge, a design methodology based on this modular assembly concept has not yet been reported.*

Response to reviewer #4's comments

General Comment: Marangoni swimmers are designed by drawing with a pen with camphor ink. Camphor release induces surface tension gradients which, in turn, causes swimmer propulsion. The authors demonstrate swimmers that can follow complex trajectories and successfully execute multistep tasks. The idea and implementation are very interesting and can inspire further design and prototyping of multifunctional swimmers. The paper is well written and organized and will be of interest to a broad readership. Several changes can be suggested before publication:

Response for General Comment: We thank the reviewer for their positive assessment of our work on Marangoni swimmers. We are pleased to hear that our demonstration of the ability of the swimmers to follow complex trajectories and execute multistep tasks was interesting and inspiring. We also appreciate the reviewer's recognition of the potential of this work to contribute to the design and prototyping of multifunctional swimmers.

We have carefully considered the changes suggested by the reviewer and incorporated them into our revised manuscript. Once again, we appreciate the reviewer's valuable feedback and are grateful for their time and effort in reviewing our manuscript.

Comment 1: In Figure 2 the legend in panel **d** need to be arranged concentration. In the same figure, clarify what the colorbars in panels e and f represents.

Response 1: We greatly appreciate your effort in examining our work and identifying errors. In the previous version of Figure 2, the color bar in panel e was represented in arbitrary units in the Abaqus simulation. We normalized the speed based on the highest speed (red) and zero speed (blue) observed in each simulation. We have updated Figure 2 to include arranged concentration and an explanation of the normalized speed.

Fig. 2. Controllable motion of pen-drawn Marangoni swimmers. *a-c* Speed of swimmer according to the drawing pattern. *a* Drawing pattern of camphor engine. *b* Speed of swimmers for different drawing patterns. *c* Average speed of swimmers between 10 s and 20 s ($n=3$). *d* Speed of swimmers under different camphor concentrations. The same drawing pattern (iii) was used in this experiment ($n=3$). *e* Different motion trajectories according to the camphor engine arrangement and estimated motion trajectories simulated by FEA. **Colorbars represent the normalized speed based on the highest speed (red) and zero-speed (blue) in each simulation.** *f* Motion programming using different concentrations of camphor inks. Scale bars: 1 cm (*e*) and 5 cm (*f*).

Comment 2: An explanation of how intermittent motion is achieved needs to be included.

Response 2: Thank you for your comment. We agree that an explanation of how intermittent motion is achieved is necessary and have included our theoretical model in Supplementary Fig. 2. However, we acknowledge that the visibility of this Supplementary Figure in the current version of the manuscript may be limited. Therefore, we have modified the manuscript to make it clearer, by referring to Supplementary Fig. 2 in the main text and providing a more visible quotation of the figure as below. We hope this addresses your concern, and we appreciate your

feedback.

The periodic motion of camphor swimmers has been investigated in previous studies^{42,44,45}. Similarly, we implemented the periodic movement of a pen-drawn Marangoni swimmer by creating an asymmetric design using camphor inks of two different concentrations (Fig. 3b, left). Our theoretical model of this periodic movement is shown in Supplementary Fig. 2.

Comment 3: Comment on the reproducibility of the results, especially when complex trajectories are considered.

Response 3: To address the reviewer's comment, we conducted additional experiments to investigate the reproducibility of the complex motion programming.

The first demonstration of complex motion programming, the star-shaped trajectory, is fairly reproducible in terms of its moving modality, which travels along the boundary while bouncing off a wall. We attempted the programming five more times, and all of the attempts succeeded in obtaining such a moving trajectory. However, the angle between vertices, which is affected by the balance between oscillation moving speed and rotation speed, was not 100% reproducible, and two out of five trials showed a 4-pointed star-shaped trajectory. We consider that this is due to the sensitivity that depends on how the pattern is drawn (e.g., thickness and lengths of lines, radius of dots), as pointed out by the reviewer, which is an intrinsic weakness of the manual hand drawing-based fabrication. We believe that this accuracy can be improved if we adopt a more accurate fabrication method, such as ink-jet based printing or lithography.

The second demonstration, the polygonal trajectory, was also highly reproducible with five successful trials out of five attempts. However, similar to the above star-shaped trajectory, the rotation angle at each vertex was not 100% reproducible, thus one of the five attempts showed a hexagonal trajectory rather than a square trajectory. We believe that this is also due to the inaccuracy caused by manual hand drawing. To be precise, we used the term “polygonal trajectory” rather than “square trajectory.”

To improve the reader's understanding, we have added more explanation in the manuscript and additional experimental data in the Supplementary Information as follows:

Manuscript

We investigated the reproducibility of these complex motion programming, as shown in Supplementary Fig. 3. First, for the programming of the star-shaped trajectory, all five attempts from the five trials were successful in drawing star-shaped trajectories that traveled along the boundary while bouncing off a wall (Supplementary Fig. 3a). However, there were some differences in the angle between each vertex depending on the attempt, which seems to be due to the inaccuracy caused by manual hand drawing. Three out of five attempts showed a five-pointed star shape, and the other two had approximately a four-pointed star shape. Secondly, the reproducibility of the polygonal trajectory was investigated as shown in Supplementary Fig. 3b. Every experiment from five attempts showed a polygonal trajectory with periodic movement, but one of them had a lack of rotation angle in the vertex, which led

to a hexagonal path, rather than a square path like the others. We believe that this inaccuracy can be improved if we adopt a more precise printing method rather than manual hand drawing.

Supplementary Information

Supplementary Fig. 3. Reproducibility of complex motion programming. **a** Reproducibility of star-shaped trajectory programming. Three out of five attempts had about five vertices during one round of the boundary of the water tank. The other two trials showed about four vertices per one round. **b** Reproducibility of polygonal trajectory programming. Four out of five attempts showed square like trajectories while the other one trial showed hexagonal trajectory.

Comment 4: It will be useful to discuss miniaturization of the technology and relevant limitations.

Response 4: Thank you for your comment. We agree that the discussion regarding miniaturization can be helpful to provide more information for authors.

Marangoni propulsion has previously been demonstrated as applicable for microswimmers in our prior work (Choi et al., Nat. Comm. 2021) and by others (Pena-Francesch et al., Nat. Comm. 2019). However, to miniaturize the Marangoni swimmer for use at the microscale, a microfabrication method such as photolithography is required to precisely pattern the Marangoni fuel on the vehicle, which potentially increases the difficulty of fabrication.

The pen-based fabrication we demonstrated in this study has two major advantages. The first is the convenience of the fabrication process from the hand drawing method, while the second is the ink-based printing, which makes it easy to process multiple inks of different concentrations on the same substrate, providing advanced programmability. The second advantage is applicable to general ink-based printing methods such as inkjet printing, not only

to manual hand drawing. Although inkjet printing is less accessible and convenient than manual hand drawing, it can offer more precise fabrication and greater potential for miniaturization while retaining the advantage of the availability to process multiple inks with different concentrations.

In summary, while this study focused on the hand drawing method for its extreme accessibility, the concept of ink-based printing of Marangoni fuel proposed in this study has applicability for more precise and miniaturized fabrication of Marangoni swimmers combined with computer-aided printing method. We have added this discussion to the manuscript as follows.

Manuscript - Discussion

Although the proposed hand drawing-based method offers exceptionally straightforward and highly accessible fabrication of Marangoni swimmers, the manual use of pen may limit the accuracy, reproducibility, and availability for miniaturization. However, the advantage of the proposed method is not just the ease of use. The ink-based printing method provides the advantage of making it easy to process different concentrations of fuel ink together on the same substrate. This enables highly advanced motion programmability. Furthermore, the ink-based fabrication of Marangoni swimmers is still compatible with conventional 2D printing technologies, such as pen-plotter and inkjet printing⁹. Combined with computer-aided design (CAD) and automated printing, more accurate, miniaturized, and mass-producible fabrication is achievable.

Comment 5: Discuss what sets the overall time of the camphor release and how the release time can be regulated.

Response 5: We appreciate your interest in the technical details of our work, and we would be happy to provide more information to clarify this aspect of our study.

From the previous studies regarding Marangoni swimmers, it has revealed that the overall time of the fuel release depends on many factors, including initial fuel loading amount (Choi et al., Nat. Comm. 2021), porosity, crosslinking density, and hydrophilicity of fuel containing-matrix (Wu et al., Nat. Comm. 2023), or impeding saturation of camphor molecules on water surface (Cheng et al., CCS Chemistry, 2019)

Fig. R1. Examples of factors affecting overall time of Marangoni fuel release from other previous research. a Lifetime depending on initial fuel loading amount. Edited from [Choi et al., Nat. Comm. 2021]. **b** Lifetime depending on matrix crosslinking density. Edited from [Lin et al., ACS Appl. Mater. Interfaces 2022].

In our study, we used several concentrations of camphor to control the power of the engine for advanced motion programmability. Higher concentrations of camphor provide longer lifetime, but the highest concentration of camphor ink used in our study (black ink, 0.64 g/mL) almost reaches the maximum solubility. Instead, we can increase the lifetime by slowing down the fuel release rate through increasing the density or hydrophobicity of the PVB matrix. We believe this can be achieved by increasing the PVB concentration in the ink or mixing other materials to increase the matrix's hydrophobicity. In this study, we used 7wt% of PVB in the ink. According to our previous study, we found that the PVB concentration can be raised up to 11 wt% while maintaining its drawability with a pen (Song et al., Sci. Adv. 2021). However, it is important to note that this approach may decrease the power of the camphor engine, which can affect the swimmer's movement. Thus, this tradeoff should be carefully considered when modifying the ink composition to increase the lifetime.

We added this discussion regarding lifetime to the Discussion section in the manuscript as below.

Manuscript - Discussion

Previous studies have revealed that the lifetime of a Marangoni swimmer is affected by various factors, including the initial fuel loading amount²⁸, fuel releasing rate^{21,46}, and the saturation of the water surface with surface tension-lowering molecules⁴⁷. Although the lifetime of our Marangoni swimmer, ranging from several minutes to 10 minutes, is shorter than those reported in recent studies, it can be prolonged by adjusting ink composition^{21,46}. Slowing down the camphor release rate by increasing the PVB concentration in ink to make the matrix denser or mixing additives to increase the matrix's hydrophobicity can be considered to prolong the swimmer's lifetime. However, it is important to note that this approach may decrease the power of the camphor engine, which can affect the swimmer's movement. Therefore, this tradeoff needs to be carefully considered when modifying the ink composition to increase the lifetime.

REVIEWERS' COMMENTS

Reviewer #2 (Remarks to the Author):

I appreciate that the authors have carefully considered all of my comments. The revisions have greatly improved the manuscript, particularly regarding reproducibility of the technique and the placement of bridging films.

Reviewer #3 (Remarks to the Author):

The authors have revised their manuscript and the revised version has become much clearer. Now, I recommend the manuscript for publication in Nature Communication.

A minor comment:

In the supplemental results, they added the description on the modeling by the other groups, and then they wrote that they performed the numerical simulation using the software ABAQUS. The manner to calculate the force and torque is clearly written, but it is not clear whether they adopt the reaction-diffusion equation for the calculation of the concentration field. This should be clarified in the text.

Reviewer #4 (Remarks to the Author):

I am satisfied with the revision and support publication of the manuscript in current form.

Response to reviewers' comments for the manuscript – “Pen-drawn Marangoni swimmer”

General Response

We appreciate the positive feedback from all reviewers. Our manuscript could be improved much based on their comments. In this letter, we address the reviewer's comments. All revised content in response to the comments is highlighted in yellow in the Supplementary Information. Text taken from the Manuscript or Supplementary Information is indicated in *italics* in this Response Letter.

Response to reviewer #2's comments

General Comment: I appreciate that the authors have carefully considered all of my comments. The revisions have greatly improved the manuscript, particularly regarding reproducibility of the technique and the placement of bridging films.

Response for General Comment: We greatly appreciate your positive feedback on the revisions we've made to our manuscript. Your comments were invaluable in improving the clarity and reproducibility of our techniques, and enhancing the section on the placement of bridging films. We're glad to know that these changes have strengthened the manuscript. Thank you for your time and effort in reviewing our work.

Response to reviewer #3's comments

General Comment: The authors have revised their manuscript and the revised version has become much clearer. Now, I recommend the manuscript for publication in Nature Communication.

Response for General Comment: We are pleased to hear that the revisions have clarified the manuscript and that you now recommend it for publication in Nature Communication. Your constructive feedback was instrumental in improving the quality and clarity of our work. We sincerely appreciate your time and effort in reviewing our manuscript.

Minor Comment: In the supplemental results, they added the description on the modeling by the other groups, and then they wrote that they performed the numerical simulation using the software ABAQUS. The manner to calculate the force and torque is clearly written, but it is not clear whether they adopt the reaction-diffusion equation for the calculation of the concentration field. This should be clarified in the text.

Response for Minor Comment: Thank you for pointing out the need for additional clarification in our supplementary information regarding the calculation of the concentration field. We apologize for any confusion caused.

In response to your comment, we have amended the supplementary information to clearly state that the reaction-diffusion equation was indeed adopted for calculating the concentration field in our numerical simulations. This clarification should provide a more comprehensive understanding of our methodology.

We appreciate your meticulous attention to detail and believe this improvement will make our methods more comprehensible for readers. Thank you once again for your constructive feedback.

Following is the part of the revised Supplementary Information.

Table. Parameters for mathematical modeling and FEA simulation

c	Camphor concentration in the bulk	[mol m ⁻³]
R	Gas constant (=8.31)	[J mol ⁻¹ K ⁻¹]
T	Absolute temperature (=297)	[K]

The driving force generated by the camphor engine, as shown below, can be derived from Reaction-diffusion and Newtonian equations (simplified version in one-dimensional space).

Based on the previous report (N. J. Suematsu et al., Langmuir 2014)⁸, when solving these two equations, the steady-state swimmers' driving force and velocity have a relationship as expressed in Equation (8). In brief, the driving force F_w is obtained by multiplying the width of the swimmer (L) by the surface pressure ($\gamma_0 - \gamma$). If we solve the reaction-diffusion equation (4) with Gibbs adsorption isotherm (7), and under the assumption that the driving force is balanced by the friction force ($F_w = \mu v_{steady}$), the steady-state driving force for the swimmers can be determined as shown in equation (8). A more detailed solving process can be found in the Suematsu's research⁸.

Reaction–diffusion equation:

$$\frac{\partial \Gamma(x,t)}{\partial t} = D \nabla^2 \Gamma(x,t) - \kappa \Gamma(x,t) + \alpha L^{-1} \delta(x - x_0) \quad (4)$$

where the delta function, δ , denotes the assumption that the camphor molecules are supplied only to the stern of the boat⁹.

Newtonian equation:

$$m \frac{d^2 x_0}{dt^2} = -\mu \frac{dx_0}{dt} + F_w = 0 \quad (5)$$

$$F_w = L(\gamma_0 - \gamma(\Gamma(x_0, t))) \quad (6)$$

Gibbs adsorption isotherm:

$$\Gamma = -\frac{c}{RT} \left(\frac{\partial \gamma}{\partial c} \right) \quad (7)$$

Driving force at steady state:

$$F_{w0} = \mu v_{steady} \sqrt{\frac{v_{steady}^2}{4kD} + 1} \quad (8)$$

Response to reviewer #4's comments

General Comment: I am satisfied with the revision and support publication of the manuscript in current form.

Response for General Comment: We are gratified to learn that you support the publication of our manuscript in its current form. Your insightful comments and recommendations have been pivotal to the development and refinement of our work. We extend our sincerest thanks for your time and effort in reviewing our manuscript.a